# Multiagent Evaluation under Incomplete Information

**Mark Rowland**[1,*]
markrowland@google.com

**Shayegan Omidshafiei**[2,*]
somidshafiei@google.com

**Karl Tuyls**[2]
karltuyls@google.com

**Julien Pérolat**[1]
perolat@google.com

**Michal Valko**[2]
valkom@deepmind.com

**Georgios Piliouras**[3]
georgios@sutd.edu.sg

**Rémi Munos**[2]
munos@google.com

[1]DeepMind London     [2]DeepMind Paris     [3] Singapore University of Technology and Design

[*]Equal contributors

## Abstract

This paper investigates the evaluation of learned multiagent strategies in the in-complete information setting, which plays a critical role in ranking and training of agents. Traditionally, researchers have relied on Elo ratings for this purpose, with recent works also using methods based on Nash equilibria. Unfortunately, Elo is unable to handle intransitive agent interactions, and other techniques are restricted to zero-sum, two-player settings or are limited by the fact that the Nash equilibrium is intractable to compute. Recently, a ranking method called $\alpha$-Rank, relying on a new graph-based game-theoretic solution concept, was shown to tractably apply to general games. However, evaluations based on Elo or $\alpha$-Rank typically assume noise-free game outcomes, despite the data often being collected from noisy sim-ulations, making this assumption unrealistic in practice. This paper investigates multiagent evaluation in the incomplete information regime, involving general-sum many-player games with noisy outcomes. We derive sample complexity guarantees required to confidently rank agents in this setting. We propose adaptive algorithms for accurate ranking, provide correctness and sample complexity guarantees, then introduce a means of connecting uncertainties in noisy match outcomes to uncer-tainties in rankings. We evaluate the performance of these approaches in several domains, including Bernoulli games, a soccer meta-game, and Kuhn poker.

## 1   Introduction

This paper investigates evaluation of learned multiagent strategies given noisy game outcomes. The Elo rating system is the predominant approach used to evaluate and rank agents that learn through, e.g., reinforcement learning [12, 35, 42, 43]. Unfortunately, the main caveat with Elo is that it cannot handle intransitive relations between interacting agents, and as such its predictive power is too restrictive to be useful in non-transitive situations (a simple example being the game of *Rock-Paper-Scissors*). Two recent empirical game-theoretic approaches are *Nash Averaging* [3] and $\alpha$-*Rank* [36]. Empirical Game Theory Analysis (EGTA) can be used to evaluate learning agents that interact in large-scale multiagent systems, as it remains largely an open question as to how such agents can be evaluated in a principled manner [36, 48, 49]. EGTA has been used to investigate this evaluation problem by deploying empirical or meta-games [37, 38, 47, 51–54]. Meta-games abstract away the

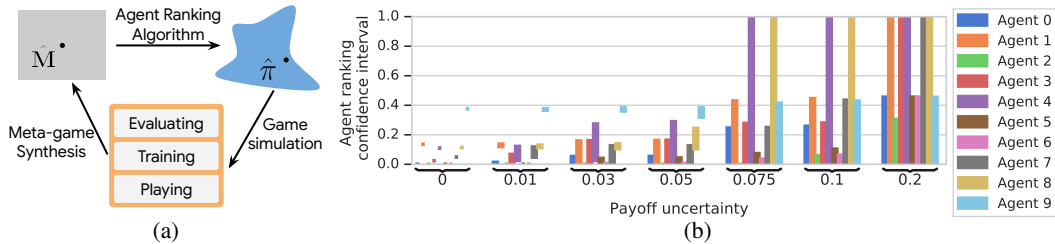

Figure 1.1: (a) Illustration of converting plausible payoff matrices consistent with an empirical estimate $\hat{\mathbf{M}}$ to empirical rankings $\hat{\boldsymbol{\pi}}$. The set of plausible payoff matrices and plausible rankings are shown, respectively, in grey and blue. (b) Ranking uncertainty vs. payoff uncertainty for a soccer meta-game involving 10 agents. Each cluster of bars shows confidence intervals over ranking weights given an observed payoff matrix with a particular uncertainty level; payoff uncertainty here corresponds to the mean confidence interval size of payoff matrix entries. This example illustrates the need for careful consideration of payoff uncertainties when computing agent rankings.

atomic decisions made in the game and instead focus on interactions of high-level agent strategies, enabling the analysis of large-scale games using game-theoretic techniques. Such games are typically constructed from large amounts of data or simulations. An *evaluation* of the meta-game then gives a means of comparing the strengths of the various agents interacting in the original game (which might, e.g., form an important part of a training pipeline [25, 26, 42]) or of selecting a final agent after training has taken place (see Fig. 1.1a).

Both Nash Averaging and $\alpha$-Rank assume noise-free (i.e., complete) information, and while $\alpha$-Rank applies to general games, Nash Averaging is restricted to 2-player zero-sum settings. Unfortunately, we can seldom expect to observe a noise-free specification of a meta-game in practice, as in large multiagent systems it is unrealistic to expect that the various agents under study will be pitted against all other agents a sufficient number of times to obtain reliable statistics about the meta-payoffs in the empirical game. While there have been prior inquiries into approximation of equilibria (e.g., Nash) using noisy observations [15, 28], few have considered evaluation or *ranking* of agents in meta-games with incomplete information [40, 53]. Consider, for instance, a meta-game based on various versions of AlphaGo and prior state-of-the-art agents (e.g., Zen) [41, 48]; the game outcomes are noisy, and due to computational budget not all agents might play against each other. These issues are compounded when the simulations required to construct the empirical meta-game are inherently expensive.

Motivated by the above issues, this paper contributes to multiagent evaluation under incomplete information. As we are interested in general games that go beyond dyadic interactions, we focus on $\alpha$-Rank. Our contributions are as follows: first, we provide sample complexity guarantees describing the number of interactions needed to confidently rank the agents in question; second, we introduce adaptive sampling algorithms for selecting agent interactions for the purposes of accurate evaluation; third, we develop means of propagating uncertainty in payoffs to uncertainty in agent rankings. These contributions enable the principled evaluation of agents in the incomplete information regime.

## 2 Preliminaries

We review here preliminaries in game theory and evaluation. See Appendix A for related work.

**Games and meta-games.** Consider a $K$-player game, where each player $k \in [K]$ has a finite set $S^k$ of pure strategies. Denote by $S = \prod_k S^k$ the space of pure strategy profiles. For each tuple $s = (s^1, \ldots, s^K) \in S$ of pure strategies, the game specifies a joint probability distribution $\nu(s)$ of payoffs to each player. The vector of expected payoffs is denoted $\mathbf{M}(s) = (\mathbf{M}^1(s), \ldots, \mathbf{M}^K(s)) \in \mathbb{R}^K$. In empirical game theory, we are often interested in analyzing interactions at a higher meta-level, wherein a strategy profile $s$ corresponds to a tuple of machine learning agents and the matrix $\mathbf{M}$ captures their expected payoffs when played against one another in some domain. Given this, the notions of 'agents' and 'strategies' are considered synonymous in this paper.

**Evaluation.** Given payoff matrix $\mathbf{M} \in (\mathbb{R}^K)^S$, a key task is to evaluate the strategies in the game. This is sometimes done in terms of a game-theoretic solution concept (e.g., Nash equilibria), but may

also consist of rankings or numerical scores for strategies. We focus particularly on the evolutionary dynamics based $\alpha$-Rank method [36], which applies to general many-player games, but also provide supplementary results for the Elo ranking system [12]. There also exist Nash-based evaluation methods, such as Nash Averaging in two-player, zero-sum settings [3, 48], but these are not more generally applicable as the Nash equilibrium is intractable to compute and select [11, 20].

The exact payoff table $\mathbf{M}$ is rarely known; instead, an empirical payoff table $\hat{\mathbf{M}}$ is typically constructed from observed agent interactions (i.e., samples from the distributions $\nu(s)$). Based on collected data, practitioners may associate a set of *plausible* payoff tables with this point estimate, either using a frequentist confidence set, or a Bayesian high posterior density region. Figure 1.1a illustrates the application of a ranking algorithm to a set of plausible payoff matrices, where rankings can then be used for evaluating, training, or prescribing strategies to play. Figure 1.1b visualizes an example demonstrating the sensitivity of computed rankings to estimated payoff uncertainties (with ranking uncertainty computed as discussed in Section 5). This example highlights the importance of propagating payoff uncertainties through to uncertainty in rankings, which can play a critical role, e.g., when allocating training resources to agents based on their respective rankings during learning.

$\alpha$**-Rank.** The Elo ranking system (reviewed in Appendix C) is designed to estimate win-loss probabilities in two-player, symmetric, constant-sum games [12]. Yet despite its widespread use for ranking [2, 19, 31, 42], Elo has no predictive power in intransitive games (e.g., Rock-Paper-Scissors) [3]. By contrast, $\alpha$-Rank is a ranking algorithm inspired by evolutionary game theory models, and applies to $K$-player, general-sum games [36]. At a high level, $\alpha$-Rank defines an irreducible Markov chain over strategy set $S$, called the *response graph* of the game [32]. The ordered masses of this Markov chain's unique invariant distribution $\boldsymbol{\pi}$ yield the strategy profile rankings. The Markov transition matrix, $\mathbf{C}$, is defined in a manner that establishes a link to a solution concept called Markov-Conley chains (MCCs). MCCs are critical for the rankings computed, as they capture agent interactions even under intransitivities and are tractably computed in general games, unlike Nash equilibria [11].

In more detail, the underlying transition matrix over $S$ is defined by $\alpha$-Rank as follows. Let $s = (s^1, \ldots, s^K) \in S$ be a pure strategy profile, and let $\sigma = (\sigma^k, s^{-k})$ be the pure strategy profile which is equal to $s$, except for player $k$, which uses strategy $\sigma^k \in S^k$ instead of $s^k$. Denote by $\eta$ the reciprocal of the total number of valid profile transitions from a given strategy profile (i.e., where only a single player deviates in her strategy), so that $\eta = (\sum_{l=1}^{K}(|S^l| - 1))^{-1}$. Let $\mathbf{C}_{s,\sigma}$ denote the transition probability from $s$ to $\sigma$, and $\mathbf{C}_{s,s}$ the self-transition probability of $s$, with each defined as:

$$\mathbf{C}_{s,\sigma} = \begin{cases} \eta \frac{1 - \exp\left(-\alpha\left(\mathbf{M}^k(\sigma) - \mathbf{M}^k(s)\right)\right)}{1 - \exp(-\alpha m(\mathbf{M}^k(\sigma) - \mathbf{M}^k(s)))} & \text{if } \mathbf{M}^k(\sigma) \neq \mathbf{M}^k(s) \\ \frac{\eta}{m} & \text{otherwise}, \end{cases} \quad \text{and} \quad \mathbf{C}_{s,s} = 1 - \sum_{\substack{k \in [K] \\ \sigma | \sigma^k \in S^k \setminus \{s^k\}}} \mathbf{C}_{s,\sigma}, \quad (1)$$

where if two strategy profiles $s$ and $s'$ differ in more than one player's strategy, then $\mathbf{C}_{s,s'} = 0$. Here $\alpha \geq 0$ and $m \in \mathbb{N}$ are parameters to be specified; the form of this transition probability is informed by particular models in evolutionary dynamics and is explained in detail by Omidshafiei et al. [36], with large values of $\alpha$ corresponding to higher *selection pressure* in the evolutionary model considered. A key remark is that the correspondence of $\alpha$-Rank to the MCC solution concept occurs in the limit of infinite $\alpha$. In practice, to ensure the irreducibility of $\mathbf{C}$ and the existence of a unique invariant distribution $\boldsymbol{\pi}$, $\alpha$ is either set to a large but finite value, or a perturbed version of $\mathbf{C}$ under the infinite-$\alpha$ limit is used. We theoretically and numerically analyze both the finite- and infinite-$\alpha$ regimes in this paper, and provide more details on $\alpha$-Rank, response graphs, and MCCs in Appendix B.

## 3 Sample complexity guarantees

This section provides sample complexity bounds, stating the number of strategy profile observations needed to obtain accurate $\alpha$-Rank rankings with high probability. We give two sample complexity results, the first for rankings in the finite-$\alpha$ regime, and the second an instance-dependent guarantee on the reconstruction of the transition matrix in the infinite-$\alpha$ regime. All proofs are in Appendix D.

**Theorem 3.1** (Finite-$\alpha$). *Suppose payoffs are bounded in the interval* $[-M_{\max}, M_{\max}]$*, and define* $L(\alpha, M_{\max}) = 2\alpha \exp(2\alpha M_{\max})$ *and* $g(\alpha, \eta, m, M_{\max}) = \eta \frac{\exp(2\alpha M_{\max}) - 1}{\exp(2\alpha m M_{\max}) - 1}$*. Let* $\varepsilon \in (0, 18 \times$

$2^{-|S|} \sum_{n=1}^{|S|-1} \binom{|S|}{n} n^{|S|}$), $\delta \in (0, 1)$. *Let $\hat{M}$ be an empirical payoff table constructed by taking $N_s$ i.i.d. interactions of each strategy profile $s \in S$. Then the invariant distribution $\hat{\pi}$ derived from the empirical payoff matrix $\hat{M}$ satisfies $\max_{s \in \prod_k S^k} |\pi(s) - \hat{\pi}(s)| \leq \varepsilon$ with probability at least $1 - \delta$, if*

$$N_s > \frac{648 M_{\max}^2 \log(2|S|K/\delta) L(\alpha, M_{\max})^2 \left( \sum_{n=1}^{|S|-1} \binom{|S|}{n} n^{|S|} \right)^2}{\varepsilon^2 g(\alpha, \eta, m, M_{\max})^2} \qquad \forall s \in S .$$

The dependence on $\delta$ and $\varepsilon$ are as expected from typical Chernoff-style bounds, though Markov chain perturbation theory introduces a dependence on the $\alpha$-Rank parameters as well, most notably $\alpha$.

**Theorem 3.2** (Infinite-$\alpha$). *Suppose all payoffs are bounded in $[-M_{\max}, M_{\max}]$, and that $\forall k \in [K]$ and $\forall s^{-k} \in S^{-k}$, we have $|M^k(\sigma, s^{-k}) - M^k(\tau, s^{-k})| \geq \Delta$ for all distinct $\sigma, \tau \in S^k$, for some $\Delta > 0$. Let $\delta > 0$. Suppose we construct an empirical payoff table $(\hat{M}^k(s) \mid k \in [K], s \in S)$ through $N_s$ i.i.d games for each strategy profile $s \in S$. Then the transition matrix $\hat{C}$ computed from payoff table $\hat{M}$ is exact (and hence all MCCs are exactly recovered) with probability at least $1 - \delta$, if*

$$N_s > 8\Delta^{-2} M_{\max}^2 \log(2|S|K/\delta) \qquad \forall s \in S .$$

A consequence of the theorem is that exact infinite-$\alpha$ rankings are recovered with probability at least $1 - \delta$. We also provide theoretical guarantees for Elo ratings in Appendix C for completeness.

# 4 Adaptive sampling-based ranking

Whilst instructive, the bounds above have limited utility as the payoff gaps that appear in them are rarely known in practice. We next introduce algorithms that compute accurate rankings with high confidence without knowledge of payoff gaps, focusing on $\alpha$-Rank due to its generality.

**Problem statement.** Fix an error tolerance $\delta > 0$. We seek an algorithm which specifies (i) a sampling scheme $\mathcal{S}$ that selects the next strategy profile $s \in S$ for which a noisy game outcome is observed, and (ii) a criterion $\mathcal{C}(\delta)$ that stops the procedure and outputs the estimated payoff table used for the infinite-$\alpha$ $\alpha$-Rank rankings, which is exactly correct with probability at least $1 - \delta$.

The assumption of infinite-$\alpha$ simplifies this task; it is sufficient for the algorithm to determine, for each $k \in [K]$ and pair of strategy profiles $(\sigma, s^{-k})$, $(\tau, s^{-k})$, whether $M^k(\sigma, s^{-k}) > M^k(\tau, s^{-k})$ or $M^k(\sigma, s^{-k}) < M^k(\tau, s^{-k})$ holds. If all such pairwise comparisons are correctly made with probability at least $1 - \delta$, the correct rankings can be computed. Note that we consider only instances for which the third possibility, $M^k(\sigma, s^{-k}) = M^k(\tau, s^{-k})$, does not hold; in such cases, it is well-known that it is impossible to design an adaptive strategy that always stops in finite time [13].

This problem can be described as a related collection of *pure exploration* bandit problems [4]; each such problem is specified by a player index $k \in [K]$ and set of two strategy profiles $\{s, (\sigma^k, s^{-k})\}$ (where $s \in S, \sigma^k \in S^k$) that differ only in player $k$; the aim is to determine whether player $k$ receives a greater payoff under strategy profile $s$ or $(\sigma^k, s^{-k})$. Each individual best-arm identification problem can be solved to the required confidence level by maintaining empirical means and a confidence bound for the payoffs concerned. Upon termination, an evaluation technique such as $\alpha$-Rank can then be run on the resulting response graph to compute the strategy profile (or agent) rankings.

## 4.1 Algorithm: ResponseGraphUCB

We introduce a high-level adaptive sampling algorithm, called ResponseGraphUCB, for computing accurate rankings in Algorithm 1. Several variants of ResponseGraphUCB are possible, depending on the choice of sampling scheme $\mathcal{S}$ and stopping criterion $\mathcal{C}(\delta)$, which we detail next.

**Sampling scheme $\mathcal{S}$.** Algorithm 1 keeps track of a list of pairwise strategy profile comparisons that $\alpha$-Rank requires, removing pairs of profiles for which we have high confidence that the empirical table is correct (according to $\mathcal{C}(\delta)$), and selecting a next strategy profile for simulation. There are several ways in which strategy profile sampling can be conducted in Algorithm 1. **Uniform (U):** A strategy profile is drawn uniformly from all those involved in an unresolved pair. **Uniform-exhaustive (UE):**

**Algorithm 1** ResponseGraphUCB($\delta, \mathcal{S}, \mathcal{C}(\delta)$)

1: Construct list $L$ of pairs of strategy profiles to compare
2: Initialize tables $\hat{\mathbf{M}}, \mathbf{N}$ to store empirical means and interaction counts
3: **while** $L$ is not empty **do**
4:      Select a strategy profile $s$ appearing in an edge in $L$ using sampling scheme $\mathcal{S}$
5:      Simulate one interaction for $s$ and update $\hat{\mathbf{M}}, \mathbf{N}$ accordingly
6:      Check whether any edges are resolved according to $\mathcal{C}(\delta)$, remove them from $L$ if so
7: **return** empirical table $\hat{\mathbf{M}}$

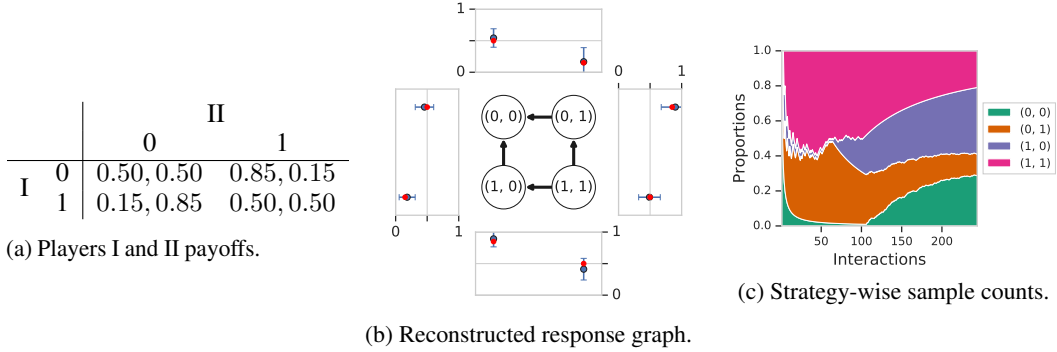

| | | II | |
|---|---|---|---|
| | | 0 | 1 |
| I | 0 | $0.50, 0.50$ | $0.85, 0.15$ |
| | 1 | $0.15, 0.85$ | $0.50, 0.50$ |

(a) Players I and II payoffs.

(b) Reconstructed response graph.

(c) Strategy-wise sample counts.

Figure 4.1: ResponseGraphUCB($\delta$ : 0.1, $\mathcal{S}$: UE, $\mathcal{C}$: UCB) run on a two-player game. (a) The payoff tables for both players. (b) Reconstructed response graph, together with final empirical payoffs and confidence intervals (in blue) and true payoffs (in red). (c) Strategy-wise sample proportions.

A pair of strategy profiles is selected uniformly from the set of unresolved pairs, and both strategy profiles are queried until the pair is resolved. **Valence-weighted (VW):** As each query of a profile informs multiple payoffs and has impacts on even greater numbers of pairwise comparisons, there may be value in first querying profiles that may resolve a large number of comparisons. Here we set the probability of sampling $s$ proportional to the squared valence of node $s$ in the graph of unresolved comparisons. **Count-weighted (CW):** The marginal impact on the width of a confidence interval for a strategy profile with relatively few queries is greater than for one with many queries, motivating preferential sampling of strategy profiles with low query count. Here, we preferentially sample the strategy profile with lowest count among all strategy profiles with unresolved comparisons.

**Stopping condition $\mathcal{C}(\delta)$.** The stopping criteria we consider are based on confidence-bound methods, with the intuition that the algorithm stops only when it has high confidence in all pairwise comparisons made. To this end, the algorithm maintains a confidence interval for each of the estimates, and judges a pairwise comparison to be resolved when the two confidence intervals concerned become disjoint. There are a variety of confidence bounds that can be maintained, depending on the specifics of the game; we consider **Hoeffding (UCB)** and **Clopper-Pearson (CP-UCB)** bounds, along with relaxed variants of each (respectively, **R-UCB** and **R-CP-UCB**); full descriptions are given in Appendix F.

We build intuition by evaluating ResponseGraphUCB($\delta$ : 0.1, $\mathcal{S}$ : UE, $\mathcal{C}$ : UCB), i.e., with a 90% confidence level, on a two-player game with payoffs shown in Fig. 4.1a; noisy payoffs are simulated as detailed in Section 6. The output is given in Fig. 4.1b; the center of this figure shows the estimated response graph, which matches the ground truth in this example. Around the response graph, mean payoff estimates and confidence bounds are shown for each player-strategy profile combination in blue; in each of the surrounding four plots, ResponseGraphUCB aims to establish which of the true payoffs (shown as red dots) is greater for the deviating player, with directed edges pointing towards estimated higher-payoff deviations. Figure 4.1b reveals that strategy profile $(0, 0)$ is the sole sink of the response graph, thus would be ranked first by $\alpha$-Rank. Each profile has been sampled a different number of times, with running averages of sampling proportions shown in Fig. 4.1c. Exploiting knowledge of game symmetry (e.g., as in Fig. 4.1a) can reduce sample complexity; see Appendix H.3.

We now show the correctness of ResponseGraphUCB and bound the number samples required for it to terminate. Our analysis depends on the choice of confidence bounds used in stopping condition

$\mathcal{C}(\delta)$; we describe the correctness proof in a manner agnostic to these details, and give a sample complexity result for the case of Hoeffding confidence bounds. See Appendix E for proofs.

**Theorem 4.1.** *The ResponseGraphUCB algorithm is correct with high probability: Given $\delta \in (0,1)$, for any particular sampling scheme there is a choice of confidence levels such that ResponseGraphUCB outputs the correct response graph with probability at least $1 - \delta$.*

**Theorem 4.2.** *The ResponseGraphUCB algorithm, using confidence parameter $\delta$ and Hoeffding confidence bounds, run on an evaluation instance with $\Delta = \min_{(s^k, s^{-k}),(\sigma^k, s^{-k})} |\mathbf{M}^k(s^k, s^{-k}) - \mathbf{M}^k(\sigma^k, s^{-k})|$ requires at most $\mathcal{O}(\Delta^{-2} \log(1/(\delta\Delta)))$ samples with probability at least $1 - 2\delta$.*

## 5 Ranking uncertainty propagation

This section considers the remaining key issue of efficiently computing uncertainty in the ranking weights, given remaining uncertainty in estimated payoffs. We assume known element-wise upper- and lower-confidence bounds $\mathbf{U}$ and $\mathbf{L}$ on the unknown true payoff table $\mathbf{M}$, e.g., as provided by ResponseGraphUCB. The task we seek to solve is, given a particular strategy profile $s \in S$ and these payoff bounds, to output the confidence interval for $\pi(s)$, the ranking weight for $s$ under the true payoff table $\mathbf{M}$; i.e., we seek $[\inf_{\mathbf{L} \leq \hat{\mathbf{M}} \leq \mathbf{U}} \pi_{\hat{\mathbf{M}}}(s), \sup_{\mathbf{L} \leq \hat{\mathbf{M}} \leq \mathbf{U}} \pi_{\hat{\mathbf{M}}}(s)]$, where $\pi_{\hat{\mathbf{M}}}$ denotes the output of infinite-$\alpha$ $\alpha$-Rank under payoffs $\hat{\mathbf{M}}$. This section proposes an efficient means of solving this task.

At the very highest level, this essentially involves finding plausible response graphs (that is, response graphs that are compatible with a payoff matrix $\hat{\mathbf{M}}$ within the confidence bounds $\mathbf{L}$ and $\mathbf{U}$) that minimize or maximize the probability $\pi(s)$ given to particular strategy profiles $s \in S$ under infinite-$\alpha$ $\alpha$-Rank. Considering the maximization case, intuitively this may involve directing as many edges adjacent to $s$ towards $s$ as possible, so as to maximize the amount of time the corresponding Markov chain spends at $s$. It is less clear intuitively what the optimal way to set the directions of edges not adjacent to $s$ should be, and how to enforce consistency with the constraints $\mathbf{L} \leq \hat{\mathbf{M}} \leq \mathbf{U}$. In fact, similar problems have been studied before in the PageRank literature for search engine optimization [7, 9, 10, 16, 24], and have been shown to be reducible to constrained dynamic programming problems.

More formally, the main idea is to convert the problem of obtaining bounds on $\boldsymbol{\pi}$ to a constrained stochastic shortest path (CSSP) policy optimization problem which optimizes *mean return time* for the strategy profile $s$ in the corresponding . In full generality, such constrained policy optimization problems are known to be NP-hard [10]. Here, we show that it is sufficient to optimize an *unconstrained* version of the $\alpha$-Rank CSSP, hence yielding a tractable problem that can be solved with standard SSP optimization routines. Details of the algorithm are provided in Appendix G; here, we provide a high-level overview of its structure, and state the main theoretical result underlying the correctness of the approach.

The first step is to convert the element-wise confidence bounds $\mathbf{L} \leq \hat{\mathbf{M}} \leq \mathbf{U}$ into a valid set of constraints on the form of the underlying response graph. Next, a reduction is used to encode the problem as policy optimization in a constrained shortest path problem (CSSP), as in the PageRank literature [10]; we denote the corresponding problem instance by $\mathrm{CSSP}(S, \mathbf{L}, \mathbf{U}, s)$. Whilst solution of CSSPs is in general hard, we note here that it is possible to remove the constraints on the problem, yielding a stochastic shortest path problem that can be solved by standard means.

**Theorem 5.1.** *The unconstrained SSP problem given by removing the action consistency constraints of $\mathrm{CSSP}(S, \mathbf{L}, \mathbf{U}, s)$ has the same optimal value as $\mathrm{CSSP}(S, \mathbf{L}, \mathbf{U}, s)$.*

See Appendix G for the proof. Thus, the general approach for finding worst-case upper and lower bounds on infinite-$\alpha$ $\alpha$-Rank ranking weights $\pi(s)$ for a given strategy profile $s \in S$ is to formulate the unconstrained SSP described above, find the optimal policy (using, e.g., linear programming, policy or value iteration), and then use the inverse relationship between mean return times and stationary distribution probabilities in recurrent Markov chains to obtain the bound on the ranking weight $\pi(s)$ as required; full details are given in Appendix G. This approach, when applied to the soccer domain described in the sequel, yields Fig. 1.1b.

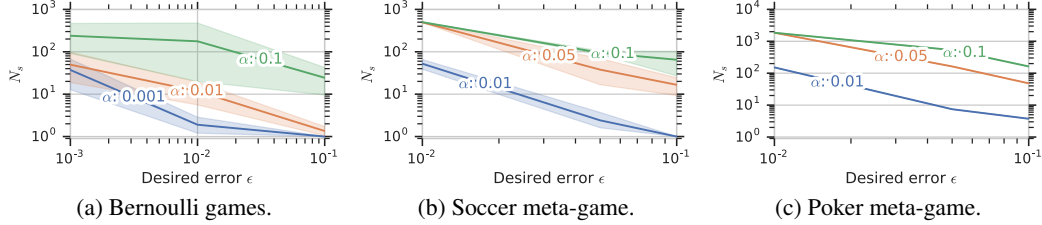

|   |   |   |
|---|---|---|
| (a) Bernoulli games. | (b) Soccer meta-game. | (c) Poker meta-game. |

Figure 6.1: Samples needed per strategy profile ($N_s$) for finite-$\alpha$ $\alpha$-Rank, without adaptive sampling.

# 6   Experiments

We consider three domains of increasing complexity, with experimental procedures detailed in Appendix H.1. First, we consider randomly-generated two-player zero-sum **Bernoulli games**, with the constraint that payoffs $\mathbf{M}^k(s, \sigma) \in [0, 1]$ cannot be too close to $0.5$ for all pairs of distinct strategies $s, \sigma \in S$ where $\sigma = (\sigma^k, s^{-k})$ (i.e., a single-player deviation from $s$). This constraint implies that we avoid games that require an exceedingly large number of interactions for the sampler to compute a reasonable estimate of the payoff table. Second, we analyze a **Soccer meta-game** with the payoffs in Liu et al. [33, Figure 2], wherein agents learn to play soccer in the MuJoCo simulation environment [46] and are evaluated against one another. This corresponds to a two-player symmetric zero-sum game with 10 agents, but with empirical (rather than randomly-generated) payoffs. Finally, we consider a **Kuhn poker meta-game** with asymmetric payoffs and 3 players with access to 3 agents each, similar to the domain analyzed in [36]; here, only $\alpha$-Rank (and not Elo) applies for evaluation due to more than two players being involved. In all domains, noisy outcomes are simulated by drawing the winning player i.i.d. from a Bernoulli($\mathbf{M}^k(s)$) distribution over payoff tables $\mathbf{M}$.

We first consider the empirical sample complexity of $\alpha$-Rank in the finite-$\alpha$ regime. Figure 6.1 visualizes the number of samples needed per strategy profile to obtain rankings given a desired invariant distribution error $\epsilon$, where $\max_{s \in \prod_k S^k} |\pi(s) - \hat{\pi}(s)| \leq \varepsilon$. As noted in Theorem 3.1, the sample complexity increases with respect to $\alpha$, with the larger soccer and poker domains requiring on the order of $1e3$ samples per strategy profile to compute reasonably accurate rankings. These results are also intuitive given the evolutionary model underlying $\alpha$-Rank, where lower $\alpha$ induces lower selection pressure, such that strategies perform almost equally well and are, thus, easier to rank.

As noted in Section 4, sample complexity and ranking error under adaptive sampling are of particular interest. To evaluate this, we consider variants of ResponseGraphUCB in Fig. 6.2, with particular focus on the UE sampler ($\mathcal{S}$: UE) for visual clarity; complete results for all combinations of $\mathcal{S}$ and $\mathcal{C}(\delta)$ are presented in Appendix Section H.2. Consider first the results for the Bernoulli games, shown in Fig. 6.2a; the top row plots the number of interactions required by ResponseGraphUCB to accurately compute the response graph given a desired error tolerance $\delta$, while the bottom row plots the number of response graph edge errors (i.e., the number of directed edges in the estimated response graph that point in the opposite direction of the ground truth graph). Notably, the CP-UCB confidence bound is guaranteed to be tighter than the Hoeffding bounds used in standard UCB, thus the former requires fewer interactions to arrive at a reasonable response graph estimate with the same confidence as the latter; this is particularly evident for the relaxed variants R-CP-UCB, which require roughly an order of magnitude fewer samples compared to the other sampling schemes, despite achieving a reasonably low response graph error rate.

Consider next the ResponseGraphUCB results given noisy outcomes for the soccer and poker meta-games, respectively in Figs. 6.2b and 6.2c. Due to the much larger strategy spaces of these games, we cap the number of samples available at $1e5$. While the results for poker are qualitatively similar to the Bernoulli games, the soccer results are notably different; in Fig. 6.2b (top), the non-relaxed samplers use the entire budget of $1e5$ interactions, which occurs due to the large strategy space cardinality. Specifically, the player-wise strategy size of 10 in the soccer dataset yields a total of 900 two-arm bandit problems to be solved by ResponseGraphUCB. We note also an interesting trend in Fig. 6.2b (bottom) for the three ResponseGraphUCB variants ($\mathcal{S}$: UE, $\mathcal{C}(\delta)$: UCB), ($\mathcal{S}$: UE, $\mathcal{C}(\delta)$: R-UCB), and ($\mathcal{S}$: UE, $\mathcal{C}(\delta)$: CP-UCB). In the low error tolerance ($\delta$) regime, the uniform-exhaustive strategy used by these three variants implies that ResponseGraphUCB spends the majority of its sampling budget observing interactions of an extremely small set of strategy profiles, and thus cannot resolve the remaining response graph edges accurately, resulting in high error. As error tolerance $\delta$ increases,

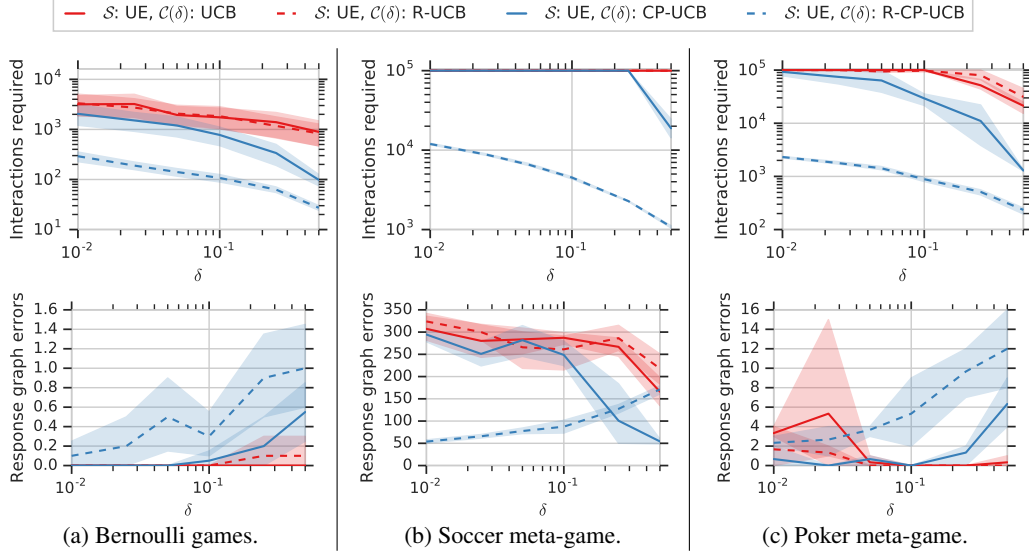

Figure 6.2: ResponseGraphUCB performance metrics versus error tolerance $\delta$ for all games. First and second rows, respectively, show the # of interactions required and response graph edge errors.

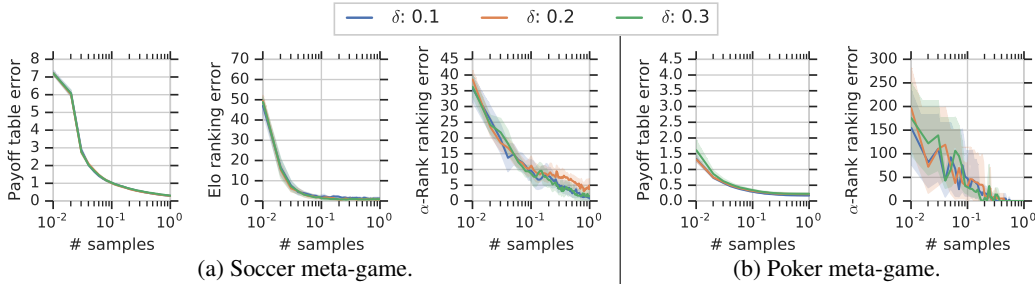

Figure 6.3: Payoff table Frobenius error and ranking errors for various ResponseGraphUCB confidence levels $\delta$. Number of samples is normalized to $[0, 1]$ on the $x$-axis.

while the probability of correct resolution of *individual* edges decreases by definition, the earlier stopping time implies that the ResponseGraphUCB allocates its budget over a larger set of strategies to observe, which subsequently lowers the *total* number of response graph errors.

Figure 6.3a visualizes the ranking errors for Elo and infinite-$\alpha$ $\alpha$-Rank given various ResponseGraphUCB error tolerances $\delta$ in the soccer domain. Ranking errors are computed using the Kendall partial metric (see Appendix H.4). Intuitively, as the estimated payoff table error decreases due to added samples, so does the ranking error for both algorithms. Figure 6.3b similarly considers the $\alpha$-Rank ranking error in the poker domain. Ranking errors again decrease gracefully as the number

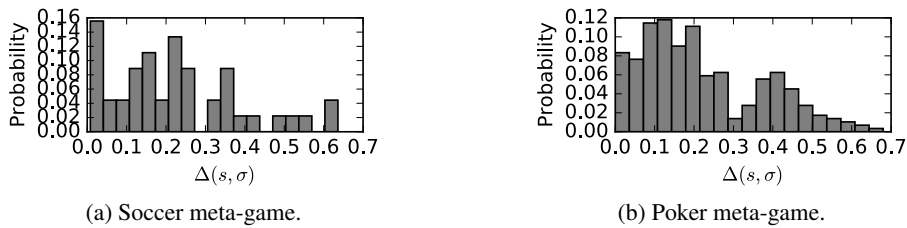

Figure 6.4: The ground truth distribution of payoff gaps for all response graph edges in the soccer and poker meta-games. We conjecture that the higher ranking variance may be explained by these gaps tending to be more heavily distributed near 0 for poker, making it difficult for ResponseGraphUCB to sufficiently capture the response graph topology given a high error tolerance $\delta$.

of samples increases. Interestingly, while errors are positively correlated with respect to the error tolerances $\delta$ for the poker meta-game, this tolerance parameter seems to have no perceivable effect on the soccer meta-game. Moreover, the poker domain results appear to be much higher variance than the soccer counterparts. To explore this further, we consider the distribution of payoff gaps, which play a key role in determining the response graph reconstruction errors. Let $\Delta(s, \sigma) = |\mathbf{M}^k(s) - \mathbf{M}^k(\sigma)|$, the payoff difference corresponding to the edge of the response graph where player $k$ deviates, causing a transition between strategy profiles $s, \sigma \in S$. Figure 6.4 plots the ground truth distribution of these gaps for all response graph edges in soccer and poker. We conjecture that the higher ranking variance may be explained by these gaps tending to be more heavily distributed near 0 for poker, making it difficult for ResponseGraphUCB to distinguish the 'winning' profile and thereby sufficiently capture the response graph topology.

Overall, these results indicate a need for careful consideration of payoff uncertainties when ranking agents, and quantify the effectiveness of the algorithms proposed for multiagent evaluation under incomplete information. We conclude by remarking that the pairing of bandit algorithms and $\alpha$-Rank seems a natural means of computing rankings in settings where, e.g., one has a limited budget for adaptively sampling match outcomes. Our use of bandit algorithms also leads to analysis which is flexible enough to be able to deal with $K$-player general-sum games. However, approaches such as collaborative filtering may also fare well in their own right. We conduct a preliminary analysis of this in Appendix H.5, specifically for the case of two-player win-loss games, leaving extensive investigation for follow-up work.

## 7 Conclusions

This paper conducted a rigorous investigation of multiagent evaluation under incomplete information. We focused particularly on $\alpha$-Rank due to its applicability to general-sum, many-player games. We provided static sample complexity bounds quantifying the number of interactions needed to confidently rank agents, then introduced several sampling algorithms that adaptively allocate samples to the agent match-ups most informative for ranking. We then analyzed the propagation of game outcome uncertainty to the final rankings computed, providing sample complexity guarantees as well as an efficient algorithm for bounding rankings given payoff table uncertainty. Evaluations were conducted on domains ranging from randomly-generated two-player games to many-player meta-games constructed from real datasets. The key insight gained by this analysis is that noise in match outcomes plays a prevalent role in determination of agent rankings. Given the recent emergence of training pipelines that rely on the evaluation of hundreds of agents pitted against each other in noisy games (e.g., Population-Based Training [25, 26]), we strongly believe that consideration of these uncertainty sources will play an increasingly important role in multiagent learning.

## Acknowledgements

We thank Daniel Hennes and Thore Graepel for extensive feedback on an earlier version of this paper, and the anonymous reviewers for their comments and suggestions to improve the paper. Georgios Piliouras acknowledges MOE AcRF Tier 2 Grant 2016-T2-1-170, grant PIE-SGP-AI-2018-01 and NRF 2018 Fellowship NRF-NRFF2018-07.

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
