[Supplementary Material · Multiagent_Evaluation_under_Incomplete_Information-Supp.pdf]

# Appendices:
# Multiagent Evaluation under Incomplete Information

We provide here supplementary material that may be of interest to the reader. Note that section and figures in the main text that are referenced here are clearly indicated via numerical counters (e.g., Fig. 1.1), whereas those in the appendix itself are indicated by alphabetical counters (e.g., Fig. H.2).

## A   Related Work

Originally, Empirical Game Theory was introduced to reduce and study the complexity of large economic problems in electronic commerce, e.g., continuous double auctions [51–53], and later it has been also applied in various other domains and settings [22, 37–39, 48]. Empirical game theoretic analysis and the effects of uncertainty in payoff tables (in the form of noisy payoff estimates and/or missing table elements) on the computation of Nash equilibria have been studied for some time [1, 15, 28, 44, 50, 55, 56], with contributions including sample complexity bounds for accurate equilibrium estimation [50], adaptive sampling algorithms [56], payoff query complexity results of computing approximate Nash equilibria in various types of games [15], and the formulation of particular varieties of equilibria robust to noisy payoffs [1]. These earlier methods are mainly based on the Nash equilibrium concept and use, amongst others, information-theoretic ideas (value of information) and regression techniques to generalize payoffs of strategy profiles. By contrast, in this paper we focus on both Elo ratings and an approach inspired by response graphs, evolutionary dynamics and Markov-Conley Chains, capturing the underlying dynamics of the multiagent interactions and providing a rating of players on their long-term behavior [36].

The Elo rating system was originally introduced to rate chess players and named after Arpad Elo [12]. It defines a measure to express the relative strength of a player, and as such has also been widely adopted in machine learning to evaluate the strength of agents or strategies [41, 42, 48]. Unfortunately, when applying Elo rating in machine learning, and multiagent learning particular, Elo is problematic: it is restricted to 2-player interactions, it is unable to capture intransitive behaviors and an Elo score can potentially be artificially inflated [3]. A Bayesian skill rating system called TrueSkill, which handles player skill uncertainties and generalized Elo rating, was introduced in Herbrich et al. [23]. For an introduction and discussion of extensions to Elo rating see, e.g., Coulom [8]. Other researchers have also introduced a method based on a fuzzy pair-wise comparison matrix that uses a cosine similarity measure for ratings, but this approach is also limited to two-player interactions [5].

Another recent work that inherently uses response graphs as its underlying dynamical model is the PSRO algorithm (Policy-Space Response Oracles) [32]. The Deep Cognitive Hierarchies model relates PSRO to cognitive hierarchies, and is equivalent to a response graph. The algorithm is essentially a generalization of the Double Oracle algorithm [34] and Fictitious Self-Play [21], iteratively computing approximate best responses to the meta-strategies of other agents.

## B   $\alpha$-Rank: Additional Background

This section provides additional background on the $\alpha$-Rank ranking algorithm.

Given match outcomes for a $K$-player game, $\alpha$-Rank computes rankings as follows:

1. Construct meta-payoff tables $\mathbf{M}^k$ for each player $k \in \{1, \ldots, K\}$ (e.g., by using the win/loss ratios for the different strategy/agent match-ups as payoffs)

2. Compute the transition matrix $\mathbf{C}$, as detailed in Section 2

3. Compute the stationary distribution, $\boldsymbol{\pi}$, of $\mathbf{C}$

4. Compute the agent rankings by ordering the masses of $\boldsymbol{\pi}$

In the transition structure outlined in Section 2 Eq. (1), the factor $(\sum_{l=1}^{K}(|S^l|-1))^{-1}$ normalizes across the different strategy profiles that $s$ may transition to, whilst the second factor represents the relative fitness of the two profiles $s$ and $\sigma$ In practice, $m \in \mathbb{N}$ is typically fixed and one considers the invariant distribution $\pi_\alpha$ as a function of the parameter $\alpha$. Figure B.1 illustrates the fixation probabilities in the $\alpha$-Rank model, for various values of $m$ and $\alpha$.

**Finite-$\alpha$ limit.** In general, the invariant distribution tends to converge as $\alpha \to \infty$, and we take $\alpha$ to be sufficiently large such that $\pi_\alpha$ has effectively converged and corresponds to the MCC solution concept.

**Infinite-$\alpha$ limit.** An alternative approach is to set $\alpha$ infinitely large, then introduce a small perturbation along every edge of the response graph, such that transitions can occur from dominated strategies to dominant ones. This perturbation enforces irreducibility of the Markov transition matrix $\mathbf{C}$, yielding a unique stationary distribution and corresponding ranking.

(a) $\alpha$-Rank population size $m = 2$.      (b) $\alpha$-Rank population size $m = 50$.

Figure B.1: Illustrations of fixation probabilities in the $\alpha$-Rank model.

|   | | II | | |
|---|---|---|---|---|
|   |   | L | C | R |
| | U | 2, 1 | 1, 2 | 0, 0 |
| I | M | 1, 2 | 2, 1 | 1, 0 |
| | D | 0, 0 | 0, 1 | 2, 2 |

(a)

(b)

(c)

Figure B.2: The response graph associated to payoffs shown in (a) is visualized in (b). (c) MCCs associated with the response graph highlighted in blue.

The **response graph** of a game is a directed graph where nodes correspond to pure strategy profiles, and directed edges if the deviating player's new strategy is a better-response. The response graph for the game specified in Fig. B.2a is illustrated in Fig. B.2b.

**Markov-Conley Chains (MCCs)** are defined as the sink strongly connected components of the response graph. The MCCs associated with the payoffs specified in Fig. B.2a are illustrated in Fig. B.2c. The stationary distribution computed by $\alpha$-Rank corresponds to a ranking of strategy profiles in the MCCs of the game response graph, indicating the average amount of time individuals in the underlying evolutionary model spend playing each strategy profile.

# C Elo Rating System: Overview and Theoretical Results

This section provides an overview of the Elo rating system, along with theoretical guarantees on the number of samples needed to construct accurate payoff matrices using Elo.

## C.1 Elo Evaluation

Consider games involving two players with shared strategy set $S^1$. Elo computes a vector $\mathbf{r} \in \mathbb{R}^{S^1}$ quantifying the strategy ratings. Let $\phi(x) = (1 + \exp(-x))^{-1}$, then the probability of $s^1 \in S^1$ beating $s^2 \in S^1$ predicted by Elo is $\mathbf{q}_{s^1, s^2}(\mathbf{r}) = \phi(\mathbf{r}_{s^1} - \mathbf{r}_{s^2})$. Consider a batch of $N$ two-player game outcomes $(s_n^1, s_n^2, u_n)_{n=1}^N$, where $\{s_n^1, s_n^2\} \in S^1$ are the player strategies and $u_n$ is the observed (noisy) payoff to player 1 in game $n$. Let $\mathbf{u} \in \mathbb{R}^N$ denote the vector of all observed payoffs, and

denote by *BatchElo* the algorithm applied to the batch of outcomes. BatchElo fits ability parameters $\mathbf{r}$ by minimizing the following objective with respect to $\mathbf{r}$:

$$L_{\text{Elo}}(\mathbf{r}; \mathbf{u}) = \sum_{n=1}^{N} -u_n \log\left(\phi(\mathbf{r}_{\mathbf{s}_n^1} - \mathbf{r}_{\mathbf{s}_n^2})\right) - (1 - u_n) \log\left(1 - \phi(\mathbf{r}_{\mathbf{s}_n^1} - \mathbf{r}_{\mathbf{s}_n^2})\right). \quad (2)$$

Ordering the elements of $\mathbf{r}$ gives the strategy rankings. Yet despite its widespread use for ranking [2, 19, 31, 42], Elo has no predictive power in intransitive games (e.g., Rock-Paper-Scissors) [3].

## C.2 Theoretical Results

In analogy with the sample complexity results for $\alpha$-Rank presented in Section 3, we give the following result on the sample complexity of Elo ranking, building on the work of Balduzzi et al. [3].

**Theorem C.1.** *Consider a symmetric, two-player win-loss game with finite strategy set $S^1$ and payoff matrix $\mathbf{M}$. Let $\mathbf{q}$ be the fitted payoffs obtained from the BatchElo model on the payoff matrix $\mathbf{M}$, and let $\hat{\mathbf{q}}$ be the fitted payoffs obtained from the BatchElo model on an empirical payoff table $\hat{\mathbf{M}}$, based on $N_{s,s'}$ interactions between each pair of strategies $s, s'$. If we take, for each pair of strategy profiles $s, s' \in S^1$, a number of interactions $N_{s,s'}$ satisfying*

$$N_{s,s'} > 0.5|S^1|^2 \varepsilon^{-2} \log(|S^1|^2/\delta). \quad (3)$$

*Then it follows that with probability at least $1 - \delta$,*

$$\left| \sum_{s'} (\mathbf{q}_{s,s'} - \hat{\mathbf{q}}_{s,s'}) \right| < \varepsilon \qquad \forall s \in S^1. \quad (4)$$

*Proof.* As in Balduzzi et al. [3, Proposition 1], we have that the row and column sums of $\hat{\mathbf{q}}, \mathbf{q}$ match those of $\hat{\mathbf{p}}, \mathbf{p}$, respectively. Thus, as a first result we obtain

$$\sum_{s'} (\mathbf{q}_{s,s'} - \hat{\mathbf{q}}_{s,s'}) = \sum_{s'} (\mathbf{q}_{s,s'} - \mathbf{p}_{s,s'}) + \sum_{s'} (\mathbf{p}_{s,s'} - \hat{\mathbf{p}}_{s,s'}) + \sum_{s'} (\hat{\mathbf{p}}_{s,s'} - \hat{\mathbf{q}}_{s,s'})$$

$$= \sum_{s'} (\mathbf{p}_{s,s'} - \hat{\mathbf{p}}_{s,s'}) \quad \forall s,$$

By analogous calculation, we obtain the following result for column sums:

$$\sum_{s'} (\mathbf{q}_{s,s'} - \hat{\mathbf{q}}_{s,s'}) = \sum_{s'} (\mathbf{p}_{s,s'} - \hat{\mathbf{p}}_{s,s'}) \quad \forall s.$$

We may now apply Hoeffding's inequality to each $\hat{\mathbf{p}}_{s,s'}$ with at least $N_{s,s'}$ samples as in the statement of the theorem, and applying a union bound yields the required inequality. $\square$

# D Proofs of results from Section 3

## D.1 Proof of Theorem 3.1

**Theorem 3.1** (Finite-$\alpha$). *Suppose payoffs are bounded in the interval $[-M_{\max}, M_{\max}]$, and define $L(\alpha, M_{\max}) = 2\alpha \exp(2\alpha M_{\max})$ and $g(\alpha, \eta, m, M_{\max}) = \eta \frac{\exp(2\alpha M_{\max}) - 1}{\exp(2\alpha m M_{\max}) - 1}$. Let $\varepsilon \in (0, 18 \times 2^{-|S|} \sum_{n=1}^{|S|-1} \binom{|S|}{n} n^{|S|})$, $\delta \in (0, 1)$. Let $\hat{\mathbf{M}}$ be an empirical payoff table constructed by taking $N_s$ i.i.d. interactions of each strategy profile $s \in S$. Then the invariant distribution $\hat{\pi}$ derived from the empirical payoff matrix $\hat{\mathbf{M}}$ satisfies $\max_{s \in \prod_k S^k} |\pi(s) - \hat{\pi}(s)| \leq \varepsilon$ with probability at least $1 - \delta$, if*

$$N_s > \frac{648 M_{\max}^2 \log(2|S|K/\delta) L(\alpha, M_{\max})^2 \left( \sum_{n=1}^{|S|-1} \binom{|S|}{n} n^{|S|} \right)^2}{\varepsilon^2 g(\alpha, \eta, m, M_{\max})^2} \qquad \forall s \in S.$$

We begin by stating and proving several preliminary results.

**Theorem D.1** (Finite-$\alpha$ confidence bounds)**.** *Suppose all payoffs are bounded in the interval* $[-M_{\max}, M_{\max}]$. *Let* $0 < \varepsilon < \frac{g(\alpha, \eta, m, M_{\max})}{2^{|S|} L(\alpha, M_{\max})}$, *and let* $\hat{\mathbf{M}}$ *be an empirical payoff table, such that*

$$\sup_{\substack{k \in [K] \\ s \in \prod_l S^l}} |\mathbf{M}^k(s^k, s^{-k}) - \hat{\mathbf{M}}^k(s^k, s^{-k})| \leq \varepsilon. \tag{5}$$

*Then, denoting the invariant distribution of the Markov chain associated with* $\hat{\mathbf{M}}$ *by* $\hat{\boldsymbol{\pi}}$, *we have*

$$\max_{s \in \prod_k S^k} |\boldsymbol{\pi}(s) - \hat{\boldsymbol{\pi}}(s)| \leq 18\varepsilon \frac{L(\alpha, M_{\max})}{g(\alpha, \eta, m, M_{\max})} \sum_{n=1}^{|S|-1} \binom{|S|}{n} n^{|S|}. \tag{6}$$

We base our proof of Theorem D.1 on the following corollary of [45, Theorem 1].

**Theorem D.2.** *Let* $q$ *be an irreducible transition kernel on a finite state space* $S$, *with invariant distribution* $\pi$. *Let* $\beta \in (0, 1/2^{|S|})$, *and let* $\hat{q}$ *be another transition kernel on* $S$. *Suppose that* $|q(t|s) - \hat{q}(t|s)| \leq \beta q(t|s)$ *for all states* $s, t \in S$. *If* $q$ *is irreducible, then* $\hat{q}$ *is irreducible, and the invariant distributions* $\pi, \hat{\pi}$ *of* $q, \hat{q}$ *satisfy*

$$|\pi(s) - \hat{\pi}(s)| \leq 18\pi(s)\beta \sum_{n=1}^{|S|-1} \binom{|S|}{n} n^{|S|},$$

*for all* $s \in S$.

We next derive several technical bounds on properties of the $\alpha$-Rank transition matrix $\mathbf{C}$, defined in (1).

**Lemma D.3.** *Suppose all payoffs are bounded in the interval* $[-M_{\max}, M_{\max}]$. *Then all non-zero elements of the transition matrix* $\mathbf{C}$ *are lower-bounded by* $g(\alpha, \eta, m, M_{\max})$.

*Proof.* Consider first an off-diagonal, non-zero element of the matrix. The transition probability is given by

$$\eta \left( \frac{1 - \exp(-\alpha x)}{1 - \exp(-\alpha m x)} \right),$$

for some $x \in [-2M_{\max}, 2M_{\max}]$, by assumption of boundedness of payoffs. This quantity is minimal for $x = -2M_{\max}$, and we hence obtain the required lower-bound. We note also that these transition probabilities are upper-bounded by taking $x = 2M_{\max}$, yielding an upper bound of $\eta \frac{1-\exp(-2\alpha M_{\max})}{1-\exp(-2\alpha m M_{\max})}$. The transition probability of a diagonal element $\mathbf{C}_{ii}$ takes the form $1 - \sum_{j \neq i} \mathbf{C}_{ij}$. There are $\eta^{-1}$ non-zero terms in the sum, each of which is upper-bounded by $\eta \frac{1-\exp(-2\alpha M_{\max})}{1-\exp(-2\alpha m M_{\max})}$. Hence, we obtain the following lower bound on the diagonal entries:

$$
\begin{aligned}
1 - \eta^{-1}\eta \frac{1 - \exp(-2\alpha M_{\max})}{1 - \exp(-2\alpha m M_{\max})} &= 1 - \frac{1 - \exp(-2\alpha M_{\max})}{1 - \exp(-2\alpha m M_{\max})} \\
&= \frac{1 - \exp(-2\alpha m M_{\max}) - 1 + \exp(-2\alpha M_{\max})}{1 - \exp(-2\alpha m M_{\max})} \\
&= \frac{\exp(2\alpha(m-1)M_{\max}) - 1}{\exp(2\alpha m M_{\max}) - 1} \\
&\geq \eta \frac{\exp(2\alpha M_{\max}) - 1}{\exp(2\alpha m M_{\max}) - 1},
\end{aligned}
$$

as required. $\square$

**Lemma D.4.** *Suppose all payoffs are bounded in the interval* $[-M_{\max}, M_{\max}]$. *All transition probabilities are Lipschitz continuous as a function of the collection of payoffs* $(\hat{\mathbf{M}}^k(s^k, s^{-k})|k \in [K], s \in \prod_l S^l)$ *under the infinity norm, with Lipschitz constant upper-bounded by* $L(\alpha, M_{\max})$.

*Proof.* We begin by considering off-diagonal, non-zero elements. The transition probability takes the form $\eta\left(\frac{1-\exp(-\alpha x)}{1-\exp(-\alpha m x)}\right)$, for some $x \in [-2M_{\max}, 2M_{\max}]$, representing the payoff difference for the pair of strategy profiles concerned. First, the Lipschitz constant of $x \mapsto \exp(-\alpha x)$ on the domain $x \in [-2M_{\max}, 2M_{\max}]$ is $\alpha\exp(2\alpha M_{\max})$. Composing this function with the function $x \mapsto \eta\frac{1-x}{1-x^m}$ yields the transition probability, and this latter function has Lipschitz constant $\eta$ on $(0, \infty)$. Therefore, the function $x \mapsto \eta\left(\frac{1-\exp(-\alpha x)}{1-\exp(-\alpha m x)}\right)$ is Lipschitz continuous on $[-2M_{\max}, 2M_{\max}]$, with Lipschitz constant upper-bounded by $\eta\alpha\exp(2\alpha M_{\max})$. Hence, the Lipschitz constant of the off-diagonal transition probabilities as a function of the payoffs under the infinity norm is upper-bounded by $2\eta\exp(2\alpha M_{\max})$. Turning our attention to the diagonal elements, we may immediately read off their Lipschitz constant as being upper-bounded by $\eta^{-1} \times 2\eta\alpha\exp(2\alpha M_{\max}) = 2\alpha\exp(2\alpha M_{\max})$, and hence the statement of the lemma follows. $\square$

We can now give the full proof of Theorem D.1.

*Proof of Theorem D.1.* By Lemma D.4, we have that all elements of the transition matrix $C$ are Lipschitz with constant $L(\alpha, M_{\max})$ with respect to the payoffs $(\mathbf{M}^k(s^k, s^{-k})|k \in [K], s \in \prod_l S^l)$ under the infinity norm. Thus, denoting the transition matrix constructed from the empirical payoff table $\hat{\mathbf{M}}$ by $\hat{\mathbf{C}}$, we have the following bound for all $i, j$:

$$|\mathbf{C}_{ij} - \hat{\mathbf{C}}_{ij}| \leq \varepsilon L(\alpha, M_{\max}).$$

Next, we have that all non-zero elements of $\mathbf{C}_{ij}$ are lower-bounded by $g(\alpha, \eta, m, M_{\max})$ by Lemma D.3, and hence we have

$$|\mathbf{C}_{ij} - \hat{\mathbf{C}}_{ij}| \leq \varepsilon L(\alpha, M_{\max}) \leq \varepsilon\frac{L(\alpha, M_{\max})}{g(\alpha, \eta, m, M_{\max})}\mathbf{C}_{ij}.$$

By assumption, the coefficient of $\mathbf{C}_{ij}$ on the right-hand side is less than $1/2^{|S|}$. We may now appeal to Theorem D.2, to obtain

$$|\pi(s) - \hat{\pi}(s)| \leq 18\pi(s)\varepsilon\frac{L(\alpha, M_{\max})}{g(\alpha, \eta, m, M_{\max})}\sum_{n=1}^{|S|-1}\binom{|S|}{n}n^{|S|},$$

for all $s \in \prod_k S^k$. Using the trivial bound $\pi(s) \leq 1$ for each $s \in \prod_k S^k$ yields the result. $\square$

With Theorem D.1 established, we can now prove Theorem 3.1.

*Proof of Theorem 3.1.* By Theorem D.1, we have that $\max_{s \in S}|\pi(s) - \hat{\pi}(s)| < \varepsilon$ is guaranteed if

$$\max_{\substack{k \in [K] \\ s \in S}}|\mathbf{M}^k(s^k, s^{-k}) - \hat{\mathbf{M}}^k(s^k, s^{-k})| < \frac{\varepsilon g(\alpha, \eta, m, M_{\max})}{18L(\alpha, M_{\max})\sum_{n=1}^{|S|-1}\binom{|S|}{n}n^{|S|}} < \frac{g(\alpha, \eta, m, M_{\max})}{2^{|S|}L(\alpha, M_{\max})}.$$

We separate this into two conditions. Firstly, from the second inequality above, we require

$$\varepsilon < \frac{18\sum_{n=1}^{|S|-1}\binom{|S|}{n}n^{|S|}}{2^{|S|}},$$

which is satisfied by assumption. Secondly, we have the condition

$$\max_{\substack{k \in [K] \\ s \in S}}|\mathbf{M}^k(s^k, s^{-k}) - \hat{\mathbf{M}}^k(s^k, s^{-k})| < \frac{\varepsilon g(\alpha, \eta, m, M_{\max})}{18L(\alpha, M_{\max})\sum_{n=1}^{|S|-1}\binom{|S|}{n}n^{|S|}}.$$

Now, write $N_s$ for the number of trials with the strategy profile $s \in S$. We will next use the following form of Hoeffding's inequality: Let $X_1, \ldots, X_N$ be i.i.d. random variables supported on $[a, b]$. Let $\varepsilon > 0$ and $\delta > 0$. Then for $N > (b - a)^2\log(2/\delta)/(2\varepsilon^2)$, we have

$\mathbb{P}\left(\left|\frac{1}{N}\sum_{n=1}^{N}X_n - \mathbb{E}\left[X_1\right]\right| > \varepsilon\right) < \delta$. Applying this form of Hoeffding's inequality to the random variable $\hat{M}^k(s^k, s^{-k})$, if we take

$$N_s > \frac{4M_{\max}^2 \log(2K|S|/\delta)}{2\left(\frac{\varepsilon g(\alpha,\eta,m,M_{\max})}{L(\alpha,M_{\max})18\sum_{n=1}^{|S|-1}\binom{|S|}{n}n^{|S|}}\right)^2} = \frac{648M_{\max}^2 \log(2K|S|/\delta)L(\alpha,M_{\max})^2\left(\sum_{n=1}^{|S|-1}\binom{|S|}{n}n^{|S|}\right)^2}{\varepsilon^2 g(\alpha,\eta,m,M_{\max})^2},$$

then

$$|\mathbf{M}^k(s^k, s^{-k}) - \hat{\mathbf{M}}^k(s^k, s^{-k})| < \frac{\varepsilon g(\alpha,\eta,m,M_{\max})}{18L(\alpha,M_{\max})\sum_{n=1}^{|S|-1}\binom{|S|}{n}n^{|S|}}$$

holds with probability at least $1 - \delta/(|S|K)$. Applying a union bound over all $k \in [K]$ and $s \in S$ then gives

$$\max_{\substack{k\in[K]\\ s\in S}} |\mathbf{M}^k(s^k, s^{-k}) - \hat{\mathbf{M}}^k(s^k, s^{-k})| < \frac{\varepsilon g(\alpha,\eta,m,M_{\max})}{18L(\alpha,M_{\max})\sum_{n=1}^{|S|-1}\binom{|S|}{n}n^{|S|}}$$

with probability at least $1 - \delta$, as required. $\qquad\square$

### D.2  Proof of Theorem 3.2

**Theorem 3.2** (Infinite-$\alpha$). *Suppose all payoffs are bounded in $[-M_{\max}, M_{\max}]$, and that $\forall k \in [K]$ and $\forall s^{-k} \in S^{-k}$, we have $|\mathbf{M}^k(\sigma, s^{-k}) - \mathbf{M}^k(\tau, s^{-k})| \geq \Delta$ for all distinct $\sigma, \tau \in S^k$, for some $\Delta > 0$. Let $\delta > 0$. Suppose we construct an empirical payoff table $(\hat{\mathbf{M}}^k(s) \mid k \in [K], s \in S)$ through $N_s$ i.i.d games for each strategy profile $s \in S$. Then the transition matrix $\hat{\mathbf{C}}$ computed from payoff table $\hat{\mathbf{M}}$ is exact (and hence all MCCs are exactly recovered) with probability at least $1 - \delta$, if*

$$N_s > 8\Delta^{-2}M_{\max}^2 \log(2|S|K/\delta) \qquad \forall s \in S.$$

We begin by stating and proving a preliminary result.

**Theorem D.5** (Infinite-$\alpha$ confidence bounds). *Suppose all payoffs are bounded in $[-M_{\max}, M_{\max}]$. Suppose that for all $k \in [K]$ and for all $s^{-k} \in S^{-k}$, we have $|\mathbf{M}^k(\sigma, s^{-k}) - \mathbf{M}^k(\tau, s^{-k})| \geq \Delta$ for all distinct $\sigma, \tau \in S^k$, for some $\Delta > 0$. Then if $|\hat{\mathbf{M}}^k(s) - \mathbf{M}^k(s)| < \Delta/2$ for all $s \in \prod_l S^l$ and all $k \in [K]$, then we have that $\hat{\mathbf{C}} = \mathbf{C}$.*

*Proof.* From the inequality $|\hat{\mathbf{M}}^k(s) - \mathbf{M}^k(s)| < \Delta/2$ for all $s \in S$, we have by the triangle inequality that $|(\hat{\mathbf{M}}^k(\sigma, s^{-k}) - \hat{\mathbf{M}}(\tau, s^{-k})) - (\mathbf{M}^k(\sigma, s^{-k}) - \mathbf{M}^k(\tau, s^{-k}))| < \Delta$ for all $k \in [K]$, $s^{-k} \in S^{-k}$, and all distinct $\sigma, \tau \in S^k$. Thus, by the assumption of the theorem, $\hat{\mathbf{M}}^k(\sigma, s^{-k}) - \hat{\mathbf{M}}(\tau, s^{-k})$ has the same sign as $\mathbf{M}^k(\sigma, s^{-k}) - \mathbf{M}^k(\tau, s^{-k})$ for all $k \in [K]$, $s^{-k} \in S^{-k}$, and all distinct $\sigma, \tau \in S^k$. It therefore follows from the expression for fixation probabilities (assuming the Fermi revision protocol), in the limit as $\alpha \to \infty$, the estimated transition probabilites $\hat{\mathbf{C}}$ exactly match the true transition probabilities $\mathbf{C}$, and hence the invariant distribution computed from the empirical payoff table matches that computed from the true payoff table. $\qquad\square$

With this result in hand, we may now prove Theorem 3.2.

*Proof of Theorem 3.2.* We use the following form of Hoeffding's inequality. Let $X_1, \ldots, X_N$ be i.i.d. random variables supported on $[a, b]$, and let $\varepsilon > 0$, $\delta > 0$. Then if $N > (b-a)^2 \log(2/\delta)/(2\varepsilon^2)$, we have $\mathbb{P}(|\frac{1}{N}\sum_{n=1}^{N}X_i - \mathbb{E}[X_1]| > \varepsilon) < \delta$. Applying this form of Hoeffding's inequality to an empirical payoff $\hat{\mathbf{M}}^k(s)$, and writing $|S| = \Pi_k |S^k|$, we obtain the result that for

$$N_s > \frac{(2M_{\max})^2 \log(2|S|K/\delta)}{2(\Delta/2)^2} = \frac{8M_{\max}^2 \log(2|S|K/\delta)}{\Delta^2},$$

we have

$$|\hat{\mathbf{M}}^k(s) - \mathbf{M}^k(s)| < \Delta/2\,,$$

with probability at least $1 - \delta/(|S|K)$. Applying a union bound over all $k \in [K]$ and all $s \in \prod_l S^l$, we obtain that if

$$N_s > \frac{8M_{\max}^2 \log(2|S|K/\delta)}{\Delta^2} \qquad \forall s \in \prod_k S^k\,,$$

then by Theorem D.5, we have that the transition matrix $\hat{\mathbf{C}}$ computed from the empirical payoff table $\hat{\mathbf{M}}$ matches the transition matrix $\mathbf{C}$ corresponding to the true payoff table $\mathbf{M}$ with probability at least $1 - \delta$. $\qquad\square$

# E  Proofs of results from Section 4

**Theorem 4.1.** *The ResponseGraphUCB algorithm is correct with high probability: Given $\delta \in (0, 1)$, for any particular sampling scheme there is a choice of confidence levels such that ResponseGraphUCB outputs the correct response graph with probability at least $1 - \delta$.*

*Proof.* We begin by introducing some notation. For a general strategy profile $s \in S$, denote the empirical estimator of $\mathbf{M}^k(s)$ after $u$ interactions by $\hat{\mathbf{M}}_u^k(s)$ and let $n_t(s)$ be the number of interactions of $s$ by time $t$, and finally let $L(\hat{\mathbf{M}}_u^k(s), \delta, u, t)$ (respectively $U(\hat{\mathbf{M}}_u^k(s), \delta, u, t)$) denote the lower (respectively upper) confidence bound for $\mathbf{M}^k(s)$ at some time index $t$ after $u$ interactions of $s$, empirical estimator $\hat{\mathbf{M}}_u^k(s)$, and confidence parameter $\delta$. We remark that in typical pure exploration problems, $t$ counts the total number of interactions; in our scenario, since we have a *collection* of best-arm identification problems, we take a separate time index $t$ for each problem, counting the number of interactions for strategy profiles concerned with each specific problem. Thus, for the best-arm identification problem concerning two strategy profiles $s, s'$, with interaction counts $n_s, n_{s'}$, we take $t = n_s + n_{s'}$.

We first apply a union bound over each best-arm identification problem:

$$\mathbb{P}(\text{Incorrect output}) \leq \sum_{(\sigma, s^{-k}), (\tau, s^{-k})} \mathbb{P}(\text{Incorrect comparison for strategy profiles } (\sigma, s^{-k}) \text{ and } (\tau, s^{-k}))\,.$$

A standard analysis can now be applied to each best-arm identification problem, following the approach of e.g., Gabillon et al. [17], Kalyanakrishnan et al. [29], Karnin et al. [30]. To reduce notational clutter, we let $s \triangleq (\sigma, s^{-k})$ and $s' \triangleq (\tau, s^{-k})$. Further, without loss of generality taking $\mathbf{M}^k(s) > \mathbf{M}^k(s')$, we have

$$\mathbb{P}(\text{Incorrect ordering of } s, s')$$
$$\leq \mathbb{P}(\exists t, u \leq t \text{ s.t. } \mathbf{M}^k(s) < L(\hat{\mathbf{M}}_u^k(s), \delta, u, t) \text{ or } \mathbf{M}^k(s') > U(\hat{\mathbf{M}}_u^k(s'), \delta, u, t))$$
$$\leq \sum_{t=1}^{\infty} \sum_{u=1}^{t} \left[ \mathbb{P}(\mathbf{M}^k(s) < L(\hat{\mathbf{M}}_u^k(s), \delta, u, t)) + \mathbb{P}(\mathbf{M}^k(s') > U(\hat{\mathbf{M}}_u^k(s'), \delta, u, t)) \right]\,.$$

Note that the above holds for *any* sampling strategy $\mathcal{S}$. We may now apply an individual concentration inequality to each of the terms appearing in the sum above, to obtain

$$\mathbb{P}(\text{Incorrect ordering of } s, s') \leq 2 \sum_{t=1}^{\infty} \sum_{u=1}^{t} f(u, \delta, (|S^k|)_k, t)\,,$$

where $f(u, \delta, (|S^k|)_k, t)$ is an upper bound on the probability of a true mean lying outside a confidence interval based on $u$ interactions at time $t$. Thus, overall we have

$$\mathbb{P}(\text{Incorrect output}) \leq \frac{|S|\sum_{k=1}^{K}(|S^k| - 1)}{2} \sum_{t=1}^{\infty} \sum_{u=1}^{t} 2f(u, \delta, (|S^k|)_k, t)\,.$$

If $f$ is chosen such that $\frac{|S|\sum_{k=1}^{K}(|S^k|-1)}{2}\sum_{t=1}^{\infty}\sum_{u=1}^{t}2f(u,\delta,(|S^k|)_k,t) \leq \delta$, then the proof of correctness is complete. It is thus sufficient to choose

$$f(u,\delta,(|S^k|)_k,t) = \frac{6\delta}{\pi^2|S|\sum_{k=1}^{K}(|S^k|-1)t^3}\,.$$

Note that this analysis has followed without prescribing the particular *form* of confidence interval used, as long as its coverage matches the required bounds above. □

**Theorem 4.2.** *The ResponseGraphUCB algorithm, using confidence parameter $\delta$ and Hoeffding confidence bounds, run on an evaluation instance with $\Delta = \min_{(s^k,s^{-k}),(\sigma^k,s^{-k})}|\mathbf{M}^k(s^k,s^{-k}) - \mathbf{M}^k(\sigma^k,s^{-k})|$ requires at most $\mathcal{O}(\Delta^{-2}\log(1/(\delta\Delta)))$ samples with probability at least $1 - 2\delta$.*

*Proof.* We adapt the approach of Even-Dar et al. [13], and use the notation introduced in the proof of Theorem 4.1 First, let $\bar{U}(\delta,u,t) = \sup_x[U(x,\delta,u,t) - x]$, and $\bar{L}(\delta,u,t) = \sup_x[x - L(x,\delta,u,t)]$. Note that if we have counts $n_s = u$ and $n_{s'} = v$ such that

$$\mathbf{M}^k(s) - \mathbf{M}^k(s') > 2\bar{U}(\delta,u,t) + 2\bar{L}(\delta,v,t)\,, \tag{7}$$

then we have

$$L(\hat{\mathbf{M}}_u^k(s),\delta,u,t) - U(\hat{\mathbf{M}}_v^k(s'),\delta,v,t) > \hat{\mathbf{M}}_u^k(s) - \bar{L}(\delta,u,t) - (\hat{\mathbf{M}}_v^k(s') + \bar{U}(\delta,v,t))$$
$$\overset{(a)}{>} \mathbf{M}^k(s) - 2\bar{L}(\delta,u,t) - \mathbf{M}^k(s') - 2\bar{U}(\delta,v,t)$$
$$> 0\,,$$

where (a) holds with probability at least $1 - 2f(u,\delta,(|S^k|)_k,t)$. Hence, with probability at least $1 - 2f(u,\delta,(|S^k|)_k,t)$ the algorithm must have terminated by this point. Thus, if $u$, $v$ and $t$ are such that (7) holds, then we have that the algorithm will have terminated with high probability. Writing $\Delta = \mathbf{M}^k(s) - \mathbf{M}^k(s')$, with all observed outcomes bounded in $[-M_{\max}, M_{\max}]$, we have

$$\bar{U}(\delta,u,t) = \bar{L}(\delta,u,t) = \sqrt{\frac{4M_{\max}^2\log(2/f(u,\delta,(|S^k|)_k,t))}{u}}\,.$$

We thus require

$$\Delta > 2\sqrt{\frac{4M_{\max}^2\log(2/f(u,\delta,(|S^k|)_k,t))}{u}} + 2\sqrt{\frac{4M_{\max}^2\log(2/f(v,\delta,(|S^k|)_k,t))}{v}}\,.$$

Taking $u = v$, and using $f(u,\delta,(|S^k|)_k,t) = \frac{6\delta}{\pi^2|S|\sum_{k=1}^{K}(|S^k|-1)t^3}$ as above, we obtain the condition

$$\Delta > 4\sqrt{\frac{4M_{\max}^2}{u}\log\left(\frac{8u^3\pi^2|S|\sum_{k=1}^{K}(|S^k|-1)}{3\delta}\right)}\,.$$

A sufficient condition for this to hold is $u = \mathcal{O}(\Delta^{-2}\log(\frac{2}{\delta\Delta}))$. Thus, if all strategy profiles $s$ have been sampled at least $\mathcal{O}(\Delta^{-2}\log(\frac{2}{\delta\Delta}))$ times, the algorithm will have terminated with probability at least $1 - 2\delta$. Up to a $\log(1/\Delta)$ factor, this matches the instance-aware bounds obtained in the previous section. □

## F  Additional material on ResponseGraphUCB

In this section, we give precise details of the form of the confidence intervals considered in the ResponseGraphUCB algorithm, described in the main paper.

**Hoeffding bounds (UCB).**   In cases where the noise distribution on strategy payoffs is known to be bounded on an interval $[a,b]$, we can use confidence bounds based on the standard Hoeffding inequality. For a confidence level $\delta$ and count index $n$, and mean estimate $\bar{x}$, this interval takes the form $(\bar{x} - \sqrt{(b-a)^2\log(2/\delta)/2n}, \bar{x} + \sqrt{(b-a)^2\log(2/\delta)/2n})$. Optionally, an additional exploration bonus based on a time index $t$, measuring the total number of samples for all strategy profiles concerned in the comparison, can be added, yielding an interval of the form $(\bar{x} - \sqrt{(b-a)^2\log(2/\delta)f(t)/n}, \bar{x} + \sqrt{(b-a)^2\log(2/\delta)f(t)/n})$, for some function $f : \mathbb{N} \to (0,\infty)$.

**Clopper-Pearson bounds (CP-UCB).** In cases where the noise distribution is known to be Bernoulli, it is possible to tighten the Hoeffding confidence interval described above, which is valid for any distribution supported on a fixed finite interval. The result is the asymmetric Clopper-Pearson confidence interval: for an empirical estimate $\overline{x}$ formed from $n$ samples, at a confidence level $\delta$, the Clopper-Pearson interval [6, 18] takes the form $(B(\delta/2; n\overline{x}, n - n\overline{x} + 1), B(1 - \delta/2; n\overline{x} + 1, n - n\overline{x})$, where $B(p; v, w)$ is the $p^{\text{th}}$ quantile of a Beta$(v, w)$ distribution.

**Relaxed variants.** As an alternative to waiting for confidence intervals to become fully disjoint before declaring an edge comparison to be resolved, we may instead stipulate that confidence intervals need only $\varepsilon$-disjoint (that is, the length of their intersection is $< \varepsilon$). This has the effect of reducing the number of samples required by the algorithm, and may be practically advantageous in instances where the noise distributions do not attain the worst case under the confidence bound (for example, low-variance noise under the Hoeffding bounds); clearly however, such an adjustment breaks any theoretical guarantees of high-probability correct comparisons.

# G   Additional material on uncertainty propagation

In this section, we provide details for the high-level approach outlined in Section 5, in particular giving more details regarding the reduction to response graph selection (in particular, selecting directions of particular edges within the response graph), and then using the PageRank-style reduction to obtain a CSSP policy optimization problem.

**Reduction to edge direction selection.** The infinite-$\alpha$ $\alpha$-Rank output is a function of the payoff table $\mathbf{M}$ only through the infinite-$\alpha$ limit of the corresponding transition matrix $\mathbf{C}$ defined in (1); this limit is determined by binary payoff comparisons for pairs of strategy profiles differing in a single strategy. We can therefore summarize the set of possible transition matrices $\mathbf{C}$ which are compatible with the payoff bounds $\mathbf{L}$ and $\mathbf{U}$ by compiling a list $E$ of response graph edges for which payoff comparisons (i.e., response graph edge directions) are uncertain under $\mathbf{L}$ and $\mathbf{U}$. Note that it may be possible to obtain even tighter confidence intervals on $\pi(s)$ by keeping track of which combinations of directed edges in $E$ are compatible with an underlying payoff table $\mathbf{M}$ itself, but by not doing so we only broaden the space of possible response graphs, and hence still obtain valid confidence bounds. The confidence interval for $\pi(s)$ could thus be obtained by computing the output of infinite-$\alpha$ $\alpha$-Rank for each transition matrix $\mathbf{C}$ that arises from all choices of edge directions for the uncertain edges in $E$. However, this set is generally exponentially large in the number of strategy profiles, and thus intractable to compute. The next step is to reduce this problem to one which is solvable using standard dynamic programming techniques to avoid this intractability.

**Reduction to CSSP policy optimization.** We now use a reduction similar to that used in the PageRank literature for optimizing stationary distribution mass [10], encoding the problem above as an SSP optimization problem. For a transition matrix $\mathbf{C}$, let $(X_t)_{t=0}^{\infty}$ denote the corresponding Markov chain over the space of strategy profiles $S$, and define the *mean return times* $\boldsymbol{\lambda} \in [0, \infty]^S$ by $\lambda(u) = \mathbb{E}\left[\inf\{t > 0 | X_t = u\} | X_0 = u\right]$, for each $u \in S$. By basic Markov chain theory, when $\mathbf{C}$ is such that $s$ is recurrent, the mass attributed to $s$ under the stationary distribution supported on the MCC containing $s$ is equal to $1/\lambda(s)$; thus, maximizing (respectively, minimizing) $\pi(s)$ over a set of transition matrices is equivalent to minimizing (respectively, maximizing) $\lambda(s)$. Define the mean hitting time of $s$ starting at $u$ for all $u \in S$ by $\boldsymbol{\varphi} \in [0, \infty]^S$, where $\varphi(u) = \mathbb{E}\left[\inf\{t > 0 | X_t = s\} | X_0 = u\right]$, whereby $\varphi(s) = \lambda(s)$; then $\boldsymbol{\varphi} = \widetilde{\mathbf{C}}\boldsymbol{\varphi} + \mathbf{1}$, where $\widetilde{\mathbf{C}}$ is the substochastic matrix given by setting the column of $\mathbf{C}$ corresponding to state $s$ to the zero vector, and $\mathbf{1} \in \mathbb{R}^S$ is the vector of ones.

Note that $\boldsymbol{\varphi}$ has the interpretation of a value function in an SSP problem, wherein the absorbing state is $s$ and all transitions before absorption incur a cost of 1. The original problem of maximizing (respectively, minimizing) $\pi(s)$ is now expressed as minimizing (respectively, maximizing) this value at state $s$ over the set of compatible transition matrices $\widetilde{\mathbf{C}}$. We turn this into a standard control problem by specifying the *action set* at each state $u \in S$ as $\mathcal{P}(\{e \in E | u \in e\})$, the powerset of the set of uncertain edges in $E$ incident to $u$; the interpretation of selecting a subset $U$ of these edges is that precisely the edges in $U$ will be selected to flow *out* of $u$; this then fully specifies the row of $\widetilde{\mathbf{C}}$ corresponding to $u$. Crucially, the action choices cannot be made independently at each state; if at state $u$, the uncertain edge between $u$ and $u'$ is chosen to flow in a particular direction, then at state $u'$ a *consistent* action must be chosen, so that the actions at both states agree on the direction

of the edge, thus leading to a *constrained* SSP optimization problem. We refer to this problem as $\mathrm{CSSP}(S, \mathbf{L}, \mathbf{U}, s)$. While general solution of CSSPs is intractable, we recall the statement of Theorem 5.1 that it is sufficient to consider the *unconstrained* version of $\mathrm{CSSP}(S, \mathbf{L}, \mathbf{U}, s)$ to recover the same optimal policy.

**Theorem 5.1.** *The unconstrained SSP problem given by removing the action consistency constraints of $\mathrm{CSSP}(S, \mathbf{L}, \mathbf{U}, s)$ has the same optimal value as $\mathrm{CSSP}(S, \mathbf{L}, \mathbf{U}, s)$.*

We conclude by restating the final statements of Section 5. In summary, the general approach for finding worst-case upper and lower bounds on infinite-$\alpha$ $\alpha$-Rank ranking weights $\pi(s)$ for a given strategy profile $s \in S$ is to formulate the unconstrained SSP described above, find the optimal policy (using, e.g., linear programming, policy or value iteration), and then use the inverse relationship between mean return times and stationary distribution probabilities in recurrent Markov chains to obtain the bound on the ranking weight $\pi(s)$ as required. In the single-population case, the SSP problem can be run with the infinite-$\alpha$ $\alpha$-Rank transition matrices; in the multi-population case, a sweep over perturbation levels in the infinite-$\alpha$ $\alpha$-Rank model can be performed, as with standard $\alpha$-Rank, to deal with the possibility of several sink strongly-connected components in the response graph.

## G.1 MCC detection

Here, we outline a straightforward algorithm for determining whether $\inf_{\mathbf{L} \leq \hat{\mathbf{M}} \leq \mathbf{U}} \pi_{\hat{\mathbf{M}}}(s) = 0$, without recourse to the full CSSP reduction described in Section 5. First, we use $\mathbf{L}$ and $\mathbf{U}$ to split the edges of the response graph into two disjoint sets $E_U$, edges for which the direction is uncertain under $\mathbf{L}$ and $\mathbf{U}$, and $E_C$, the edges with certain direction. We then construct the set $F_A \subseteq S$ of *forced ancestors* of $s$; that is, the set of strategy profiles that can reach $s$ using a path of edges contained in $E_C$, including $s$ itself. We also define the set $F_D \subseteq S$ of *forced descendents* of $s$; that is, the set of strategy profiles that can be reached from $s$ using a path of edges in $E_C$, including $s$ itself. If $F_D \not\subseteq F_A$, then $s$ can clearly be made to lie outside an MCC by setting all edges in $E_U$ incident to $F_D \setminus F_A$ to be directed *into* $F_D$. Then there are no edges directed out of $F_D \setminus F_A$, so this set contains at least one MCC. There also exists a path from $s$ to $F_D \setminus F_A$, and hence $s$ cannot lie in an MCC, so $\inf_{\mathbf{L} \leq \hat{\mathbf{M}} \leq \mathbf{U}} \pi_{\hat{\mathbf{M}}}(s) = 0$. If, on the other hand, $F_D \subseteq F_A$, we may set all uncertain edges between $F_A$ and its complement to be directed away from $F_A$. We then iteratively compute two sets: $F_{A,\mathrm{out}}$, the set of profiles in $F_A$ for which there exists a path out of $F_A$, and its complement $F_A \setminus F_{A,\mathrm{out}}$. Any uncertain edges between these two sets are then set to be directed towards $F_{A,\mathrm{out}}$, and the sets are then recomputed. This procedure terminates when either $F_{A,\mathrm{out}} = F_A$, or there are no uncertain edges left between $F_A$ and $F_{A,\mathrm{out}}$. If at this point there is no path from $s$ out of $F_A$, we conclude that $s$ must lie in an MCC, and so $\inf_{\mathbf{L} \leq \hat{\mathbf{M}} \leq \mathbf{U}} \pi_{\hat{\mathbf{M}}}(s) > 0$, whilst if such a path does exist, then $s$ does not lie in an MCC, so $\inf_{\mathbf{L} \leq \hat{\mathbf{M}} \leq \mathbf{U}} \pi_{\hat{\mathbf{M}}}(s) = 0$.

## G.2 Proof of Theorem 5.1

**Theorem 5.1.** *The unconstrained SSP problem given by removing the action consistency constraints of $\mathrm{CSSP}(S, \mathbf{L}, \mathbf{U}, s)$ has the same optimal value as $\mathrm{CSSP}(S, \mathbf{L}, \mathbf{U}, s)$.*

*Proof.* Let $\widetilde{\mathbf{C}}$ be the substochastic matrix associated with the optimal *unconstrained* policy, and suppose there are two action choices that are inconsistent; that is, there exist strategy profiles $u$ and $v$ differing only in index $k$, such that either (i) at state $u$, the edge direction is chosen to be $u \to v$, and at state $v$, the edge direction is chosen to be $v \to u$; or (ii) at state $u$, the edge direction is chosen to be $v \to u$, and at state $v$, the edge direction is chosen to be $u \to v$. We show that in either case, there is a policy without this inconsistency that achieves at least as good a value of the objective as the inconsistent policy.

We consider first case (i). Let $\varphi$ be the associated expected costs under the inconsistent optimal policy, and suppose without loss of generality that $\varphi(v) \geq \varphi(u)$. Let $\widetilde{\mathbf{D}}$ be the substochastic matrix obtained by adjusting the action at state $u$ so that the edge direction between $u$ and $v$ is $v \to u$, consistent with the action choice at $v$. Denote the expected costs under this new transition matrix $\widetilde{\mathbf{D}}$

by $\mu$. We can compare $\varphi$ and $\mu$ via the following calculation. By definition, we have $\varphi = \widetilde{\mathbf{C}}\varphi + \mathbf{1}$ and $\mu = \widetilde{\mathbf{D}}\mu + \mathbf{1}$. Thus, we compute

$$
\begin{aligned}
\varphi - \mu &= (\widetilde{\mathbf{C}}\varphi + \mathbf{1}) - (\widetilde{\mathbf{D}}\mu + \mathbf{1}) \\
&= \widetilde{\mathbf{C}}\varphi - \widetilde{\mathbf{D}}\mu \\
&= \widetilde{\mathbf{C}}\varphi - \widetilde{\mathbf{D}}\varphi + \widetilde{\mathbf{D}}\varphi - \widetilde{\mathbf{D}}\mu \\
&= (\widetilde{\mathbf{C}} - \widetilde{\mathbf{D}})\varphi + \widetilde{\mathbf{C}}(\varphi - \mu) \\
\implies \varphi - \mu &= (\mathbf{I} - \widetilde{\mathbf{D}})^{-1}(\widetilde{\mathbf{C}} - \widetilde{\mathbf{D}})\varphi .
\end{aligned}
$$

In this final line, we assume that $\mathbf{I} - \widetilde{\mathbf{D}}$ is invertible. If it is not, then it follows that $\widetilde{\mathbf{D}}$ is a strictly stochastic matrix, thus corresponding to a policy in which no edges flow into $s$. From this we immediately deduce that the minimal value of $\varphi(s)$ is $\infty$; hence, we may assume $\mathbf{I} - \widetilde{\mathbf{D}}$ is invertible in what follows. Assume for now that $s \notin \{u, v\}$. Now note that $\widetilde{\mathbf{C}}$ and $\widetilde{\mathbf{D}}$ differ only in two elements: $(u, u)$, and $(u, v)$, and thus the vector $(\widetilde{\mathbf{C}} - \widetilde{\mathbf{D}})\varphi$ has a particularly simple form; all coordinates are 0, except coordinate $u$, which is equal to $\eta(\varphi(v) - \varphi(u)) \geq 0$. Finally, observe that all entries of $(\mathbf{I} - \widetilde{\mathbf{D}})^{-1} = \sum_{k=0}^{\infty} \widetilde{\mathbf{D}}^k$ are non-negative, and hence we obtain the element-wise inequality $\varphi - \mu \geq 0$, proving that the policy associated with $\widetilde{\mathbf{D}}$ is at least as good as $\widetilde{\mathbf{C}}$, as required. The argument is entirely analogous in case (ii), and when one of the strategies concerned is $s$ itself. Thus, the proof is complete. $\square$

## H Additional empirical details and results

### H.1 Experimental procedures and reproducibility

We detail the experimental procedures here.

The results shown in Fig. 1.1b are generated by computing the upper and lower payoff bounds given a mean payoff matrix and confidence interval size for each entry, then running the procedure outlined in Section 5.

As Fig. 4.1 shows an intuition-building example of the ResponseGraphUCB outputs, it was computed by first constructing the payoff table specified in the figure, then running ResponseGraphUCB with the parameters specified in the caption. The algorithm was then run until termination, with the strategy-wise sample counts in Fig. 4.1 computed using running averages.

The finite-$\alpha$ $\alpha$-Rank results in Fig. 6.1 for every combination of $\alpha$ and $\epsilon$ are computed using 20, 5, and 5 independent trials, respectively, for the Bernoulli, soccer, and poker meta-games. The same number of trials applies for every combination of $\delta$ and ResponseGraphUCB in Fig. 6.3.

The ranking results shown in Fig. 6.3 are computed for 10 independent trials for each game and each $\delta$.

The parameters swept in our plots are the error tolerance, $\delta$, and desired error $\epsilon$. The range of values used for sweeps is indicated in the respective plots in Section 6, with end points chosen such that sweeps capture both the high-accuracy/high-sample complexity and low-accuracy/low-sample complexity regimes.

The sample mean is used as the central tendency estimator in plots, with variation indicated as the 95% confidence interval that is the default setting used in the `Seaborn` visualization library that generates our plots. No data was excluded and no other preprocessing was conducted to generate these plots.

No special computing infrastructure is necessary for running ResponseGraphUCB, nor for reproducing our plots; we used local workstations for our experiments.

## H.2 Full comparison plots

As noted in Section 4, sample complexity and ranking error under adaptive sampling are of particular interest. To evaluate this, we consider all variants of ResponseGraphUCB in Fig. H.1.

Figure H.1: ResponseGraphUCB performance metrics versus error tolerance $\delta$ for all games. First and second rows, respectively, show the # of interactions required and response graph edge errors.

## H.3 Exploiting knowledge of symmetry in games

(a) Reconstructed response graph.

(b) Strategy-wise sample counts.

Figure H.2: ResponseGraphUCB ($\delta = 0.1$, $\mathcal{S}$ =UE, $\mathcal{C}$ =UCB) evaluated on the game with payoff tables shown in Fig. 4.1a, with knowledge of game symmetry exploited to reduce the total number of samples needed from 244 to 20 and sampling conducted for only a single strategy profile, $(0, 1)$.

**Symmetric games.** Let $\text{Sym}_K$ denote the symmetric group of degree $K$ over all players. A game is said to be symmetric if for any permutation $\rho \in \text{Sym}_K$, strategy profile $(s^1, \ldots, s^K) \in S$ and index $k \in [K]$, we have $\mathbf{M}^k(s^1, \ldots, s^K) = \mathbf{M}^{\rho(k)}(s^{\rho(1)}, \ldots, s^{\rho(K)})$.

**Exploiting symmetry in ResponseGraphUCB.** Knowledge of the symmetric constant-sum nature of a game (e.g., Fig. 4.1a) can significantly reduce sample complexity in ResponseGraphUCB: this knowledge implies that payoffs for all symmetric strategy profiles are known a priori (e.g., payoffs for $(0, 0)$ and $(1, 1)$ are 0.5 in this example); moreover, each observed outcome for a strategy profile $(s^1, \ldots, s^K)$ yields a 'free' observation of $\mathbf{M}^{\rho(k)}(s^{\rho(1)}, \ldots, s^{\rho(K)})$ for all permutations $\rho \in \text{Sym}_K$, strategy profiles $(s^1, \ldots, s^K) \in S$, and players $k \in [K]$. For the example in Fig. 4.1a, the symmetry-exploiting variant of the algorithm is able to reconstruct the true underlying response graph using only 20 samples of a single strategy profile $(0, 1)$.

In Fig. H.2, we evaluate ResponseGraphUCB on the game shown in Fig. 4.1a, this time exploiting the knowledge of game symmetry as discussed in Section 4.1. Note that Figs. H.2a and H.2b should be compared, respectively, with Figs. 4.1b and 4.1c in the main paper. Confidence bounds corresponding to the symmetry-exploiting sampler (Fig. H.2a) are guaranteed to be tighter than the non-exploiting sampler (Fig. 4.1b), and so typically we can expect the former to require fewer interactions to arrive at a ranking conclusion with the same confidence as the latter (under the condition that the payoffs really are symmetric, as is the case in win/loss two-player games). This is observed in this particular example, where Fig. 4.1b took 244 interactions to solve, while Fig. H.2a took only 20 samples of a single strategy profile $(0, 1)$ to correctly reconstruct the response graph.

## H.4 Kendall's distance for partial rankings

We use Kendall's distance for partial rankings [14] when comparing two rankings, $r$ and $\hat{r}$ (e.g., as done in Fig. 6.3)

Consider a pair of partial strategy rankings $r$ and $\hat{r}$ (i.e., wherein tied rankings are allowed). Define a fixed parameter $p$. The Kendall distance with penalty parameter $p$ is defined,

$$K(r, \hat{r}; p) = \sum_{\{i,j\} \in [|S|]} \bar{K}_{i,j}(r, \hat{r}; p),$$

where $\bar{K}_{i,j}(r, \hat{r}; p)$ is:

- 0 when $i, j$ are in distinct buckets in both $r, \hat{r}$, but in the same order (e.g., $r_i > r_j$ and $\hat{r}_i > \hat{r}_j$)
- 1 when $i, j$ are in distinct buckets in both $r, \hat{r}$, but in the reverse order (e.g., $r_i > r_j$ and $\hat{r}_i < \hat{r}_j$)

- 0 when $i, j$ are in the same bucket in both $r$ and $\hat{r}$

- $p$ when $i, j$ are in the same bucket in one of $r$ or $\hat{r}$, but different buckets in the other.

It can be shown that Kendall's distance is a metric when $p \in [0.5, 1]$. We use $p = 0.5$ in our experiments.

### H.5 Preliminary experiments on collaborative filtering-based approaches

The pairing of bandit algorithms and $\alpha$-Rank seems a natural means of computing rankings in settings where, e.g., one has a limited budget for adaptively sampling match outcomes. Our use of bandit algorithms also leads to analysis which is flexible enough to be able to deal with $K$-player general-sum games. However, approaches such as collaborative filtering may also fare well in their own right. We conduct a preliminary analysis of this in here, specifically for the case of two-player win-loss games.

For such games, the meta-payoff table is given by a matrix $\mathbf{M}$ with all entries lying in $(0, 1)$ (encoding loss as payoff $0$ and win as payoff $1$). Taking a matrix completion approach, we might attempt to reconstruct a low-rank approximation of the payoff table from an incomplete list of (possible noisy) payoffs, and then run $\alpha$-Rank on the reconstructed payoffs. Possible candidates for the low-rank structure include: (i) the payoff matrix itself; (ii) the *logit matrix* $\mathbf{L}_{ij} = \log(\mathbf{M}_{ij}/(1 - \mathbf{M}_{ij}))$; and (iii) the *odds matrix* $\mathbf{O}_{ij} = \exp(\mathbf{L}_{ij})$. In particular, Balduzzi et al. [3] make an argument for the (approximate) low-rank structure of the logit matrix in many applications of interest.

Figure H.3: Ranking errors (Kendall's distance w.r.t. ground truth) from completion of, respectively, the sparse payoffs, logits, and odds matrices for Soccer dataset. 20 trials per combo of assumed matrix ranks and observation rates/density.

We conduct preliminary experiments on this in Fig. H.3, implementing matrix completion calculations via Alternating Minimization [27]. We compare here the resulting $\alpha$-Rank errors for the three reconstruction approaches for the Soccer meta-game. We sweep across the observation rates of payoff matrix entries and the matrix rank assumed in the reconstruction. Interestingly, conducting low-rank approximation on the logits (as opposed to the odds) matrix generally yields the lowest ranking error. Overall, the bandit-based approach may be more suitable when one can afford to play all strategy profiles at least once, whereas matrix completion is perhaps more so when this is not feasible. These results, we believe, warrant additional study of the performance of related alternative approaches in future work.