[Reviews · NeurIPS 2019]

Reviewer 1



This paper investigates the evaluation of multiagent strategies in the incomplete information and general-sum setting. The primary algorithm to be analyzed is alpha-rank, which is a ranking algorithm based on the stationary distribution of a markov chain with states defined over all strategy profiles. Since payoff tables M are typically estimated empirically, the authors provide sample complexity bounds on the number of (uniformly distributed) observations of each strategy profile to be observed for the resultant stationary distribution to be close to the true stationary distribution. The authors propose an adaptive sampling strategy based on confidence intervals over each pair of strategy profiles and analyze its sample complexity. The paper also shows how to propagate uncertainties in M to uncertainty in the ranking weights that alpha-rank yield. Lastly, the authors empirically evaluate their sample complexity bounds (in both adaptive and non-adaptive setting), and the performance of ResponseGraphUCB on 3 different problem domains. Clarity: The paper is generally well written and relatively easy to follow. The problem is well motivated and most key ideas are sufficiently well explained in the main text, with the bulk of proofs being deferred to the appendix. I do have a few minor criticisms. (1) Figure 6.2 should be presented better. Currently, there are 16 plots per graph and it is extremely difficult to make sense of each plot, not to mention confidence bounds. Since not all of these 16 plots are considered at any time, consider reporting only results which yield interesting insights in the main paper, while leaving the rest of the plots to the appendix. (2) Section 5 may be too heavy in content for readers unfamiliar with CSSPs, Markov chain theory, or “standard control problems”. In my opinion, it may be more effective to provide a high level idea of the technique used and leave precise mathematical details to the appendix. Quality: Experiments are conducted on reasonable domains and are fairly extensive. I did not verify all the derivations in the appendix, but the ideas presented in section 4 appear sound. Significance: This paper tackles the problem of estimating payoff values (and ranking weights) from empirical simulations and provides sample complexity bounds. This is a good theoretical contribution to an area which is rapidly gaining popularity, and should be of interest to researchers in many communities. Other comments: (3) Is Theorem 4.1 agnostic to the chosen sampling scheme? Empirically, how do the different sampling strategies compare with each other? (4) With regard to Figure 6.3, the authors mention that errors are positively correlated with error tolerances. This appears to be the case for the poker meta-game, and delta has virtually no visible effect on the soccer meta-game. (5) Again with regard to Figure 6.3: In both meta-games, payoff errors decreased gracefully with increasing samples, which is expected. However, ranking errors were significantly more variable in poker as compared to soccer. Could the authors provide some insight as to why this was so? (6) For Bernoulli games, it was stated that payoffs were constrained such that they would not be too close to 0.5. Is this related to theorems 3.1, 3.2, and 4.2? If this is so, would it not be a good opportunity to empirically evaluate the tightness of bounds in 4.2? ============================= Post-rebuttal. The authors have addressed my concerns satisfactorily. I maintain that the paper is a good submission given some minor edits and the addition of some of the responses in the authors' rebuttal.

Reviewer 2



Originality: - The paper is a novel combination of two known techniques - AlphaRank and UCB. It also casually adds in approaches from the PageRank literature, albeit they are not the central contributions. - The most important parts are formulating the usage of UCB in computing the accurate rankings, running rigorous experiments across a variety of different schemes, and giving algorithmic bounds for the approach. It does each of these well, with the supplementary materials providing well-written proofs for each of the sections. Quality: - This is clearly a complete work with the claims presented well and sufficiently addressed. - My main question comes in this section and accompanies what I think is the only fault of the paper; It doesn't address when or if UCB is actually the best approach to this problem. Yes, it's a good idea to use it because the formulation as a bandit problem makes a lot of sense. But is it actually the ideal in all circumstances? And when does that break down? For example, I could see collaborative filtering approaches working well in some settings, especially ones that are able to actively address which strategy pairing to query next. In what scenarios will the UCB approach be better than that one? Clarity: - The paper is well written, albeit the supplementary material is absolutely necessary to make sense of it properly as key parts are relegated to that area. - I think the paper as written is mostly reproducible. I didn't understand how the reduction to CSSP works, likely because I am not familiar with that literature, and appreciated the further explanation in the appendix. Significance: - These results are important. They build upon a recent paper that was also important as it reinvigorated a field and pushed it into the open. This paper then takes that one and shores up problems it had wrt noise in the evaluations. - I could see practitioners not using this as it adds a lot of complexity to alpha-rank, however it is definitely a line of work that will be explored further and this was/is one of the avenues of exploration.

Reviewer 3



This paper studies the practical algorithm that computes alpha-ranking for a pool of agents in MARL. The setting is broad, covering multiplayer general-sum game, while Elo-score only intends for two-player and cannot deal with intransitive agents. The proposed algorithm is based on graph bandit, which is useful when it is intractable to construct a full payoff matrix based on empirical game outcomes. The proposed algorithm would lead to an efficient way to leverage the game outcomes and assign the computation budget for what games should be carried out. Finally, it is beyond a pure evaluation method, in the sense that any MARL based on player-pool-sampling can potentially benefit from the proposed algorithm.

[Author Response · NeurIPS 2019]

We thank the reviewers for their positive feedback and will make the suggested improvements in the paper. Given the
extra page allowed for accepted papers, we will include the discussions and experiments resulting from this feedback.
We kindly hope the reviewers take this into account when finalizing their scores. Responses below are ordered w.r.t. the
reviewer comments (e.g., **R2.3** refers to Reviewer 2's 3rd comment).

**R1.1** **Figure 6.2 presentation:** Agreed, Fig. 6.2's presentation will be improved. We thank the re-
viewer for the suggestions. We will move extraneous plots from Fig. 6.2 to the appendix and increase
the plot sizes for readability. We will do the same for other figures where applicable. **R1.2 Sec. 5,**
**move CSSP details to the appendix:** Agreed, per the reviewer's suggestion, we will expound on the
higher-level details & intuitions in the section itself, and move the technical details to the appendix.

**R1.3 Theorem 4.1 & sampling strategy comparisons:** Indeed, Theorem 4.1 is agnostic
of the chosen sampling scheme; while we note this in the text preceding the theorem,
we will update the theorem statement itself to make this property explicit. Note that
Fig 6.2 visualizes empirical differences between the different sampling strategies. Overall,
CP-UCB seems to attain best empirical performance by a small margin, which can
be explained due to the Bernoulli meta-game outcomes (which are win/loss in nature).

(a) Soccer meta-game

**R1.4 & 1.5 Fig. 6.3:** Good point, we will correct the wording regarding the positive
correlation of errors and tolerances here. Regarding the higher variance of Poker results,
let us consider the distribution of payoffs gaps, which play a key role in determining
response graph reconstruction errors. Let $\Delta(s, \sigma) = |\mathbf{M}^k(s) - \mathbf{M}^k(\sigma)|$, the payoff
difference corresponding to the edge of the response graph where player $k$ deviates,
causing a transition between strategy profiles $s, \sigma \in S$ (see paper lines 91–92 for precise
definitions). Figure R1 plots the ground truth distribution of these gaps for all response
graph edges in Soccer & Poker. The higher ranking variance may be explained by these
gaps tending to be more heavily distributed near 0 for Poker. **R1.6 Bernoulli game**
**payoffs:** We agree that it would be interesting to evaluate bound tightness, and plan to investigate this in future work.

(b) Poker meta-game

Figure R1: Distribution of payoff gaps $\Delta(s, \sigma)$.

**R2.1 & 2.2 Alternative approaches, e.g., collaborative filtering:** We thank the reviewer for the interesting and
important question regarding alternative approaches. The pairing of bandit algorithms and $\alpha$-Rank is a natural means of
computing rankings in settings where, e.g., one has a limited budget for adaptively sampling match outcomes. Our use
of bandit algorithms also leads to analysis which is flexible enough to be able to deal with $K$-player general-sum games.
However, approaches such as collaborative filtering may indeed fare well in their own right. We provide discussion of
one such application below, specifically for the case of two-player win-loss games.

For such games, the meta-payoff table is given by a matrix $\mathbf{M}$ with all entries lying in $(0, 1)$ (encoding loss as payoff 0
and win as payoff 1). Taking a matrix completion approach, we might attempt to reconstruct a low-rank approximation
of the payoff table from an incomplete list of (possible noisy) payoffs, and then run $\alpha$-Rank on the reconstructed
payoffs. Possible candidates for the low-rank structure include: (i) the payoff matrix itself; (ii) the *logit matrix*
$\mathbf{L}_{ij} = \log(\mathbf{M}_{ij}/(1 - \mathbf{M}_{ij}))$; and (iii) the *odds matrix* $\mathbf{O}_{ij} = \exp(\mathbf{L}_{ij})$. In particular, Balduzzi et al. (2018) make an
argument for the (approximate) low-rank structure of the logit matrix in many applications of interest.

Per the reviewer's suggestion, we have now conducted preliminary experiments on this in Fig. R2, implementing matrix
completion calculations via Alternating Minimization (Jain et al., 2013). We compare here the resulting $\alpha$-Rank errors
for the three reconstruction approaches for the Soccer meta-game. We sweep across the observation rates of payoff
matrix entries and the matrix rank assumed in the reconstruction. Interestingly, conducting low-rank approximation on
the logits (as opposed to the odds) matrix generally yields the lowest ranking error. Overall, the bandit-based approach
may be more suitable when one can afford to play all strategy profiles at least once, whereas matrix completion is
perhaps more so when this is not feasible. We will append these discussions and results to the paper.

Figure R2: Ranking errors (Kendall's distance w.r.t. ground truth) from completion of, respectively, the sparse payoffs,
logits, and odds matrices for Soccer dataset. 20 trials per combo of assumed matrix ranks and observation rates/density.

**R3.1** We thank the reviewer for the positive feedback. Indeed, the bounds in Sec. 3 are not directly related to the final
algorithm, in contrast to the bound in Sec. 4; we also plan to investigate the possibility of tightening the former bounds.

**References:** Balduzzi, D., Tuyls, K., Perolat, J., & Graepel, T. (2018). *Re-evaluating evaluation.*
Jain, P., Netrapalli, P., & Sanghavi, S. (2013). *Low-rank matrix completion using alternating minimization.*


[Meta-Review · NeurIPS 2019]

This paper provides sample complexity bounds which relate the number of observations in strategy profiles needed to obtain good alpha-Rank rankings. Overall the authors found the paper interesting, well-written and complete. Please make sure you address all of the reviewers concerns in your final version.